

**1** **Size-resolved chemical composition, effective density, and optical**

**2** **properties of biomass burning particles**

**3** Jinghao Zhai[1], Xiaohui Lu[1], Ling Li[1], Qi Zhang[1,2], Ci Zhang[1], Hong Chen[1], Xin

**4** Yang[1]*, Jianmin Chen[1]

**5** *[1]Shanghai Key Laboratory of Atmospheric Particle Pollution and Prevention, Department of*

**6** *Environmental Science and Engineering, Fudan University, Shanghai 200433, China*

**7** *[2]Department of Environmental Toxicology, University of California, Davis, California 95616,*

**8** *United States*

**9** Correspondence to: Xin Yang (yangxin@fudan.edu.cn)

**11** **Abstract.** Biomass burning aerosol has important impact on the global radiative

**12** budget. A better understanding of the mixing state and chemical composition of

**13** biomass burning particles relative to their optical properties is the goal of a number of

**14** current studies. In this work, effective density, chemical composition, and optical

**15** properties of rice straw burning particles in the size range of 50-400 nm were

**16** measured using a suite of comprehensive methods. A Differential Mobility Analyzer

**17** (DMA)-Aerosol Particle Mass analyzer (APM)-Condensation Particle Counter (CPC)

**18** system offered detailed information on the effective density as well as mixing state of

**19** size-resolved particles. The effective density and chemical composition of individual

**20** particles were characterized with a DMA in-line with a Single Particle Aerosol Mass

**21** Spectrometer (SPAMS), simultaneously. The multiple modes observed in the

**22** size-resolved particle effective density distribution indicated size-dependent external

**23** mixing of black carbon (BC), organic carbon (OC) and potassium salts in particles.

**24** Particles of 50 nm had the smallest effective density (1.16 g/cm$^3$), due to a relative

**25** large proportion of aggregate BC. The average effective densities of 100-400 nm

**26** particles ranged from 1.35-1.51 g/cm$^3$ with OC and inorganic salts as dominant

**27** components. Both density distribution and single-particle mass spectrometry showed

**28** more complex mixing states in larger particles. Upon heating, the separation of the

**29** effective density distribution modes testified the existence of less volatile BC or soot

**30** and potassium salts. Size-resolved optical properties of biomass burning particles

**31** were measured by the Cavity Attenuated Phase Shift spectroscopy (CAPS, λ=450 &

**32** 530 nm). The single scattering albedo (SSA) showed the lowest value for 50 nm

**33** particles (0.741±0.007 & 0.889±0.006) because of larger proportion of BC content.

**34** Brown carbon played an important role for the SSA of 100-400 nm particles. The

**35** Ångström absorption exponent (AAE) values for all particles were above 1.6,

**36** indicating the significant presence of brown carbon. Though freshly emitted, the light

**37** absorption enhancement ($E_{abs}$) was observed for particles larger than 200 nm.

**38** Concurrent measurements in our work provide a basis for discussing the

**39** physicochemical properties of biomass burning aerosol and its effects on global

**40** climate and atmospheric environment.

**41**

**42** **1 Introduction**



Biomass burning is a significant source of trace gases and aerosol particles (Andreae and Merlet, 2001). Biomass burning particles affect climate by both absorbing and scattering solar radiation (Chand et al., 2009) and serve as cloud condensation nuclei which would modify cloud microphysical properties (Petters et al., 2009). In addition, biomass burning particles have considerable impacts on air quality, regional visibility, and human health (Naeher et al., 2007; Park et al., 2006). Global annual emissions of black carbon (BC) and organic carbon (OC) aerosols are estimated to be ~8 and 33.9 Tg yr$^{-1}$ while open burning contributes approximately 42% for BC and 74% for OC (Bond et al., 2004). Along with rapid economic development and increase in agricultural activities, emissions from agricultural residue combustion in China have drawn extensive attention. The total amount of straws from open burning in China is estimated to be ~140 Tg yr$^{-1}$ (Cao et al., 2008). Controlling biomass burning emissions is thus necessary to improve air quality in China.

Mixing state, composition, and morphology of particles can influence their radiative properties. An enhancement of BC forcing by up to a factor of 2.9 is estimated by models when BC is internally mixed with other components compared with externally mixed scenarios (Jacobson 2001). Quantitative assessment of light absorption properties of biomass burning particles has drawn widely attention as the radiative impacts of these particles are both affected by the strongly absorbing BC and organic matter (Chakrabarty et al., 2010). The co-emission of BC and OC can lead to internally mixed particles, in which the OC coating can enhance particle absorption through lensing effects (Bond et al., 2006; Schnaiter et al., 2005). For internally mixed BC, the assumption of a void-free BC sphere with a material density of 1.8 g/cm$^3$ can lead to overestimations of the shell/core ratio and absorption enhancement by ~13 and ~17%, respectively (Zhang et al., 2016). In addition to absorption enhancement by internal mixing, some organic matter containing specific functional groups (e.g. nitrated/polycyclic aromatics, phenols) can itself absorb radiation in the low-visible and UV wavelengths (Hoffer et al., 2006; Jacobson, 1999) and is referred to as brown carbon (BrC). As biomass burning is a significant source of BrC, the optical properties of biomass burning particles need to be further understood. Field works have been conducted to measure the light absorption enhancement by particle coatings in different areas (Chan et al., 2011; Nakayama et al., 2014). The degree to which particles absorb light depends on their composition, shape, and mixing state. Researches on chemical composition and mixing state of biomass burning particles have been done by our group members previously (Huo et al., 2016; Zhai et al., 2015). However, it remains unclear how mixing states and chemical composition of biomass burning particles influence their morphology and optical properties.

Particles emitted from biomass burning are generally composed of a mixture of spherical and non-spherical particles and chain aggregates (Martins et al., 1998). Scanning electron microscopy (SEM) as well as transmission electron microscopy (TEM) are common techniques widely used to investigate the morphology of biomass burning particles (China et al., 2013; Giordano et al., 2015). However, these methods are unable to provide continuous "on-line" information and suffer from limitations arising from primary particle overlap, screening effects, and cluster anisotropy



(Wentzel et al., 2003). Effective density is a good predictor for the complex properties
of biomass burning particles (Pitz et al., 2008) and is often used to convert particle
size distributions into mass loading (Tang and Munkelwitz, 1994). Variations in
particle effective density can be used to follow compositional transformations during
chemical reactions (Katrib et al., 2005). Online measurements which provide
real-time monitoring of particle effective density variation have been developed.
Kelly and McMurry (1992) developed a density measurement technique based on the
selection of a monodisperse aerosol with a Differential Mobility Analyzer (DMA)
followed by classification according to aerodynamic diameter with an impactor.
McMurry et al. (2002) reported a technique to determine size-resolved effective
density based on using an Aerosol Particle Mass analyzer (APM) to measure the mass
of particles that had been classified according to electrical mobility by a DMA. The
DMA-APM method has been applied extensively in field studies as well as laboratory
experiments (Hu et al., 2012; Barone et al., 2011). However, few measurements of the
effective density of biomass burning particles have been done due to the lack of
accompanying on-line chemical information.
Mixing state of individual particle can be very different caused by the chemical
composition, aging degree, etc., which greatly influence the morphology and optical
property of particles. Thus, distinction among particles might be covered by bulk
measurements. Single particle mass spectrometry techniques have been utilized to
measure the chemical composition, size, density, and shape of individual particles.
Spencer et al. (2007) utilized a DMA-ultrafine aerosol time-of-flight mass
spectrometer (UF-ATOFMS) system to detect the effective density and chemical
composition simultaneously of ambient aerosol at single-particle level. The
comprehensive information about single particles could help to elucidate the
morphology, mixing state, and sphericity of biomass burning particles.
The chemical composition, morphology, and optical properties of particles are
usually interrelated. However, given that biomass burning particles is a complex
mixture of organic and inorganic species, including strongly light-absorbing BC and
BrC, size-resolved or even single particle level information on the morphology,
chemical composition, and optical properties of biomass burning particles are needed
to help get a macroscopic understanding of the relationships. In this study, laboratory
experiments were conducted on aerosols from combusting rice straws, a main source
of biomass burning particles in Southern China. The size-resolved effective density of
biomass burning particles was measured by two different methods. One was based on
a DMA-APM-Condensation Particle Counter (CPC) system. For the other method, the
mobility size-selected particles by a DMA were transported into a Single Particle
Aerosol Mass Spectrometer (SPAMS), where the vacuum aerodynamic diameter and
chemical composition of individual particles were measured. Size-resolved optical
properties of biomass burning particles were also measured by Cavity Attenuated
Phase Shift spectroscopy (CAPS). A thermodenuder (TD) was used to help analyze
the mixing state of particles by removing the volatile compounds and leaving behind
the less volatile species based on the vaporization temperature of materials. The
purpose of our study was to add physicochemical knowledge regarding biomass





burning particles which is an important aerosol source globally.

## 2 Experiments

### 2.1 Laboratory-made biomass burning particles

Rice straw, a typical type of crop residue in Southern China, was taken as the
representative biomass burning material in our experiment. The self-designed
combustion setup was introduced in previous work (Huo et al., 2016). Briefly, the rice
straws collected in rural residential area in Shanghai were dehydrated for 24 h at
$100^{o}C$ in an oven prior to combustion. Five replicate tests of straw-burning were
conducted for each experiment. For each test, ~50g of dried rice straws were burned
in a combustion stove at a flaming condition. The emitted smoke was introduced into
a 4.5 $m^3$ (in volume) chamber with a flowrate of 50 L/min. Ambient air was
introduced though a high efficiency particulate air filters to maintain the ambient
pressure. The particles in chamber were then introduced into the measurement system
through a silica gel type diffusion drier (shown in Figure 1).

### 2.2 Single particle mass spectrometry

A Single Particle Aerosol Mass Spectrometer (SPAMS) (Hexin Analytical Instrument
Co., Ltd) was deployed to examine the aerosol chemical composition and
aerodynamic diameter at single-particle level. Detailed information on the SPAMS
has been described elsewhere (Li et al., 2011). Briefly, particles in the size range of
0.2-2.0 μm are first drawn into the vacuum through an Aerodynamic Focusing Lens.
Each particle is accelerated to a size-dependent aerodynamic velocity which is
calculated based on two orthogonally oriented continuous lasers (Nd: YAG, 532 nm).
The two lasers fixed at a 6 cm distance and the delay of the scatter light is collected
by two photomultiplier tubes (PMT). When a particle arrives at the ion source region,
a pulsed desorption/ionization laser (Qswitched Nd: YAG, 266 nm) is triggered. Ions
are recorded by a bipolar time-of-flight spectrometer, which records both positive and
negative mass spectra for each single particle. In this work, the power of desorption
/ionization laser was set to ~0.6 mJ per pulse. The aerodynamic diameter
measurement is calibrated with curves generated by monodisperse polystyrene latex
spheres (Nanosphere Size Standards, Duke Scientific Corp.) with known diameters
(0.2-2.0 μm).
All single particle mass spectra acquired were converted to a list of peaks at each
m/z by setting a minimum signal threshold of 30 arbitrary units above the baseline
with TSI MS-Analyze software. The resulting peak lists together with other SPAMS
data were imported into YAADA (version 2.11, www.yaada.org), a software toolkit
for single-particle data analysis written in Matlab (version R2011b). In this work, a
total of 10220 biomass burning particles were chemically analyzed according to their
positive and negative ion spectra, accounting for about 48 % of all sized particles.
According to the similarities of the mass-to-charge ratio and peak intensity, the
biomass burning particles were classified using an adaptive resonance theory-based
clustering method (ART-2a) (Song et al., 1999). Parameters for ART-2a used in this
work such as vigilance factor, learning rate, and iterations were 0.85, 0.05, and 20,
respectively. The particle clusters resulting from ART-2a were then grouped into 6





particle types based on the mass spectral patterns and chemical similarities. The name
of a particle type reflects the dominant chemical species.
**2.3 Effective density measurements**
**2.3.1 Theoretical calculation and methods**
Particle density ($\rho_p$) is referenced to the volume equivalent diameter ($d_{ve}$) which is
defined as the diameter of a spherical particle with the same volume as the particle
under consideration. Particle density can be derived as follows, where $m_p$ is the
particle mass:
$$\rho_p = \frac{m_p}{\frac{\pi}{6}d_{ve}^3} \tag{1}$$

When particles are not spherical, the "effective density", not necessarily a true
measurement of particle density is derived. Various definitions of effective density are
provided in the literature, and a review of these definitions is given by DeCarlo et al.
(2004). Different definitions may aim to present different values for a given particle.
It is important to understand the derivation, calculation, and measurement for one
method of particle effective density.
(1) DMA-APM-CPC system

The effective density of a particle can be calculated by combining mobility and
mass measurements under the assumption that the particle is spherical, thus its
physical diameter equals to the electrical mobility diameter ($d_m$) measured by a DMA.
The effective density ($\rho_{eff}^{I}$) can be calculated by the following equation:
$$\rho_{eff}^{I} = \frac{m_p}{\frac{\pi}{6}d_m^3} \tag{2}$$

where $m_p$ stands for particle mass obtained by an APM. In our work, we selected
biomass burning particles with mobility diameters of 50 nm, 100 nm, 200 nm, and
400 nm and determined their effective density using the DMA-APM-CPC system.
(2) DMA-SPAMS system

Another approach of deriving effective density is through a combination of
mobility and aerodynamic measurements. Simultaneously measuring the particle
electrical mobility diameter ($d_m$) by DMA and the vacuum aerodynamic diameter ($d_{va}$)
by SPAMS allows for the determination of particle effective density ($\rho_{eff}^{II}$) by the
following equation:
$$\rho_{eff}^{II} = \frac{d_{va}}{d_m}\rho_0 \tag{3}$$

where $\rho_0$ is the standard density (1.0 g/cm$^3$). In this study, since particles smaller
than 200 nm may not scatter sufficient light to be detected by SPAMS and the number
concentration of biomass burning particles above 400 nm was low (shown in Figure
S1), we selected 200 nm and 400 nm particles by DMA and then introduced them into
SPAMS.
(3) Shape factor calculation

The shape of particles can influence the optical properties and can reflect the
mixing state of particles to some degree. It is possible to extract the shape information





based on the measurements above.

The relationship between the volume equivalent diameter ($d_{ve}$) and mobility
diameter ($d_m$) is shown in the following equation:
$$\frac{d_m}{C_c(d_m)} = \frac{d_{ve}\chi}{C_c(d_{ve})}$$
(4)

where $\chi$ is the shape factor, the ratio of the resistance force on the nonspherical
particle to the resistance force on its volume equivalent sphere (Hinds, 1999). The $\chi$
value equals 1 for spherical particles and is greater than 1 for nonspherical/irregular
particles.

$Cc$ is the Cunningham Slip Correction Factor parameterized as:

$$Cc(d) = 1 + \frac{2\lambda}{d}[\alpha + \beta\exp{(-\gamma\frac{d}{2\lambda})}]$$
(5)

where $d$ is the particle diameter ($d_m$ $or$ $d_{ve}$) and $\lambda$ is the mean free path of gas
molecules. The empirical constants $\alpha$, $\beta$, and $\gamma$ are 1.142, 0.558, and 0.999
respectively (Allen and Raabe, 1985).

The vacuum aerodynamic diameter ($d_{va}$) is related to the volume equivalent
diameter ($d_{ve}$) by:
$$d_{va} = \frac{\rho_p}{\rho_0}\frac{d_{ve}}{\chi}$$
(6)

As the measurements of mobility and aerodynamic diameters are readily
available, we assumed the error was in the particle mass measurement if the measured
$\rho_{eff}^{II}$ is used to replace $\rho_{eff}^{I}$ in Equation (2) (Decarlo et al., 2004). With assumed
particle density ($\rho_p$) and known particle mass ($m_p$) measured by an APM, a calculated
$d_{ve}$ could be obtained using Equation (1). Here we assumed $\rho_p$ equals to 1 which
would be canceled out later. Using the same $d_{ve}$ and for any shape factor ($\chi$), a
calculated $d_m$ and $d_{va}$ was obtained by Equation (4) and (6), respectively. Thus, $\rho_{eff}^{II}$
could be obtained by the calculated $d_m$ and $d_{va}$ and an estimated $m_p$ was calculated by
replacing $\rho_{eff}^{I}$ by $\rho_{eff}^{II}$ in Equation (2). We then calculated the ratio of the
estimated $m_p$ to the exact $m_p$ as a function of $d_m$ and $\chi$ (shown in Figure S5, discussed
in Section 3.1.5).
**2.3.2 Instruments**
The size distribution of biomass burning particles was detected by a Scanning
Mobility Particle Sizer (SMPS) consisting of a Differential Mobility Analyzer (DMA,
Model 3080, TSI Inc.) and a Condensation Particle Counter (CPC, Model 3775, TSI
Inc.). An Aerosol Particle Mass analyzer (APM, Model 3601, Kanomax Inc.) was
used to classify aerosol particles according to their mass-to-charge ratio. The detailed
information of the APM classification principle was previously reviewed by Tajima et
al. (2011). Briefly, particles were size-selected by DMA after being charged with a
Kr85 neutralizer. Particles with a known size were then introduced into APM. When
the radial electrical and centrifugal forces were in balance, particles passed through
the rotating cylinders to CPC. Mass distribution was obtained by voltage scanning and



particle counting.

**2.4 Optical measurements**

Cavity Attenuated Phase Shift (CAPS) spectroscopy (Shoreline Science Research Inc.)
was used to determine the particle extinction and scattering coefficient. Detailed
information on the CAPS is available in Onasch et al. (2015). Briefly, a square-wave
modulated light-emitting diode (LED) is transmitted through an optical cavity cell. A
sample cell incorporating two high reflectivity mirrors (R~0.9999) with a vacuum
photodiode detector (Hamamatsu R645) centers at the wavelength of the LED. The
particle extinction coefficient [$b_{ext}(\lambda)$] can be obtained from the changes in the phase
shift of the distorted waveform of the LED. An integrating nephelometer using a 10
cm diameter integrating sphere is operated to measure the scattering coefficient [$b_{scat}$
$(\lambda)$]. Particles are illuminated by the collimated light beam which has measured the
extinction. The scattered light of particles is collected at all angels by the integrating
sphere. A PMT (H7828-01, Hamamatsu) with a high voltage power supply and an
amplifier records the scattered light. In this work, we used two CAPSs with the LED
light sources at wavelength of 450 nm and 530 nm to detect the optical properties of
biomass burning particles, respectively.

**2.5 Thermodenuder**

A thermodenuder (TD, Model 3065, TSI Inc.) was utilized to separate volatile and
less volatile species of biomass burning particles at specific temperatures. The TD
consists of a 40 cm long desorber section and a 70 cm long adsorption tube. The
sample can be heated up to 400 °C in the desorber section while we selected 150 °C
and 300 °C in this work. The adsorption tube is surrounded by an annular bed of
activated carbon which adsorbs the evaporated gas-phase compounds, leaving behind
the less volatile fractions. With a flowrate of 0.6 L/min, the residence time of particles
in the TD heating section was approximately 9 s in this work.
The particle number fractions after heating do not necessarily represent the
actual number fractions before heating as some of the particles can evaporate
completely. Besides, particle loss could be produced both in the TD heating and
adsorption section due to thermophoretic forces and diffusion, respectively (Philippin
et al., 2004). On account of the quantitative measurements of optical properties,
particle loss could lead to the underestimate of $b_{ext}$ and $b_{scat}$.
Sodium chloride (NaCl) aerosol produced by a single-jet atomizer (Model 9302,
TSI Inc.) was used to determine the transport efficiency ($\eta$) in TD. The transport
efficiencies of NaCl of different electric mobility diameters selected by DMA ($d_m$: 50,
100, 200, and 400 nm) at a range of temperatures ($T_i$: 20, 150, and 300 °C) are shown
in Figure S2. In TD, $\eta$ decreased with increasing $T_i$ and decreasing $d_m$, which was
consistent with the result in Philippin et al. (2004). The measured $\eta$ were used to
correct the particle number concentration in the calculation of optical properties.

**3 Result and discussion**

**3.1 Size-resolved effective density**

**3.1.1 Effective density from DMA-APM-CPC measurements ($\rho_{eff}^{I}$)**




The effective density of particles, measured using the DMA-APM-CPC system ($\rho_{eff}^{I}$),
provided useful information on the mixing state of particles. A Gaussian model was
applied to determine the effective densities of the biomass burning particles selected
by DMA (shown in Figure 2). The density distribution of 50 nm ($d_m$) particles showed
a single peak profile with a peak value of 1.17 g/cm$^3$ (Table S1). Two possible factors
could be inferred from this feature: a nearly-monodisperse aerosol effective density
distribution or a juncture of two models with very close peak values. Biomass burning
particles contain highly agglomerated structures like soot (Martins et al., 1998).
Although the material density of black carbon (BC) is ~1.8 g/cm$^3$ (Malm et al., 2005),
fresh BC particles with an aggregate structure can have an effective density less than
1.0 g/cm$^3$. The density of organic matter varies in the range of 1.2-2.0 g/cm$^3$
depending on sources (Hand et al., 2010; Turpin and Lim, 2001). Since particles of 50
nm have the possibility of containing organic matter rather than BC alone, the
apparent single-peak density distribution of these particles was more likely due to the
combination of two models representing BC and organic particles respectively (as the
dash lines shown in Figure 2). The thermal desorption method can help to explain the
mixing state of 50 nm particles which will be discussed in Section 3.1.3.

The density distribution of 100 nm particles exhibited a peak at 1.45 g/cm$^3$,

which suggests that these particles were dominated by organic matter. However,
less-massive composition with effective density of 0.9-1.1 g/cm$^3$ was also obtained
for 100 nm particles. This range is identical with the density of fresh BC with
aggregate structure. The bimodal distribution of the density profile of 100 nm
particles suggests that BC was partly externally mixed with other components in
ultrafine particles from biomass burning emissions. Similar result has been found by
Lack et al. (2012) and Adachi et al. (2011). The external mixing of BC and organic
particulate matter was evident in the density distribution of 200 nm particles as well
(Figure 2). For 400 nm particles, besides a dominant density mode at 1.34 g/cm$^3$, a
relative weak mode with effective density of 1.92 g/cm$^3$ was observed. Previous
studies have shown that potassium chloride crystals, which have a material density of
~ 1.99 g/cm$^3$ (Lide, 2008), were observed in the TEM of fresh biomass burning
particles (Li et al., 2015). Thus, we estimate that the mode at 1.92 g/cm$^3$ was
associated with KCl, and possibly KSO$_4$ and KNO$_3$, and that these crystalline species
were more likely externally mixed with organic matter in biomass burning particles.

Though freshly emitted, biomass burning particles can be coated by secondary

species, such as ammonium nitrate and ammonium sulfate, pronouncedly in a very
short period (Leskinen et al., 2007). The bulk densities of ammonium nitrate and
ammonium sulfate are ~1.75 g/cm$^3$. The differences in the peak values of the
dominant mode observed for 50-400 nm particles are associated with the composition
and morphology of particles. Different proportions of the same material can lead to
differences in particle effective density. The dominant modes for biomass burning
particles in the size range of 50-400 nm (Figure 2) could be a mixture of similar
composition (organic matter and secondary species) but different proportions.
Detailed information and discussion about the particle composition can be found in





Section 3.2.

### 3.1.2 Effective density from DMA-SPAMS measurements ($\rho_{eff}^{II}$)

The vacuum aerodynamic size distributions of 200 nm and 400 nm electrical mobility
selected biomass burning particles are shown in Figure 3. The dominant mode for the
200 nm mobility selected particles was 280 nm in vacuum aerodynamic diameter with
an effective density ($\rho_{eff}^{II}$) of 1.40 g/cm$^3$ and a second mode at 360 nm ($d_{va}$) with an
effective density of 1.80 g/cm$^3$. This is quite consistent with the result from the
DMA-APM-CPC method. The less intense mode at 520 nm ($d_{va}$) should be due to
doubly charged particles (Spencer et al., 2007). For 400 nm mobility selected particles,
the dominant mode in aerodynamic diameter was 540 nm with an effective density of
1.35 g/cm$^3$. Since the less massive modes at 660 nm and 840 nm were not in the
deviation range of doubly charged particles, these two modes were singly charged
particles with effective density of 1.65 and 2.10 g/cm$^3$, respectively. The
single-particle level chemical composition of biomass burning particles will be
discussed below.

### 3.1.3 Thermal denuded particle effective density

The average density distributions of 50-400 nm ($d_m$) biomass burning particles after
heating at 150$^o$C and 300$^o$C, respectively, are shown in Figure 2. After heating by TD,
the bi-model density distributions of biomass burning particles became more
pronounced. At 150$^o$C, the effective density mode with peak at ~1.0 g/cm$^3$ protruded
for the whole size range of 50-400 nm particles. The separation of the peaks helped to
testify that the less volatile BC or soot with effective density of ~1.0 g/cm$^3$ was partly
externally mixed with other compositions. The dominant density peak values for 50,
100, 200, and 400 nm particles at 150$^o$C were 1.64-1.80 g/cm$^3$. Li et al. (2016)
reported that the density of organic matter vaporized at 150$^o$C was 0.61-0.90 g/cm$^3$.
The increase of the dominant density peak value (1.34-1.45 g/cm$^3$ for unheated vs.
1.64-1.80 g/cm$^3$ for 150$^o$C heated) could be due to the volatilization of organics with
small effective density. The dominant density peak values of 50-400 nm particles at
300$^o$C were 1.75-2.04 g/cm$^3$. The volatilization temperatures of ammonium nitrate
and ammonium sulfate were reported to be ~48-89$^o$C and ~178-205$^o$C, respectively
(Johnson et al., 2004a; Johnson et al., 2004b). Thus, the fractions of ammonium
nitrate and ammonium sulfate should be small at 300$^o$C. The increase of dominant
density peak value for 50-400 nm biomass burning particles upon heating could be
due to the vaporization of volatile organics with small effective density and secondary
inorganic species such as $NH_4NO_3$ and $(NH_4)_2SO_4$ with density of ~1.75 g/cm$^3$. Upon
heating, the density mode of KCl and partly $K_2SO_4$ at ~2.0 g/cm$^3$ was ambiguous as
the dominant mode shifted right and overlapped the KCl mode (dash lines shown in
Figure 2). However, at 300$^o$C, the dominant mode of 400 nm particles was at 2.05
g/cm$^3$ which fitted the density of potassium salts, indicating the main material of 400
nm heated (~800 nm unheated, detected by a tandem DMAs, discussed in Section
3.3.3) biomass burning particles should be potassium salts.

With heating by TD, the aerodynamic size distributions of 200 nm and 400 nm





electrical mobility size-selected biomass burning particles at 300 $^{o}$C are shown in
Figure S3. The increase of $\rho_{eff}^{II}$ upon heating was consistent with that of $\rho_{eff}^{I}$.
**3.1.4 Comparison of $\rho_{eff}^{I}$ and $\rho_{eff}^{II}$**
Figure S4 summarizes that the average effective densities ($\rho_{eff}^{I}$ & $\rho_{eff}^{II}$) of biomass
burning particles that were size-selected at 6 different mobility diameters. Note that
the density distributions of the 300 nm and 350 nm ($d_m$) particles are not contained in
Figure 2 since they were similar to those of the 200 nm and 400 nm ($d_m$) particles.
The 50 nm biomass burning particles had the lowest effective density of $1.15\pm0.23$
g/cm$^3$ which could be due to the aggregate structure of black carbon. Compared with
the 50 nm ($d_m$) particles, the effective density of 100 nm particles was larger ($1.45\pm$
$0.15$ g/cm$^3$). Since the sampling limitation of SPAMS was 200 nm, $\rho_{eff}^{II}$ was derived
only for particles in the size range of 200-400 nm ($d_m$). Overall, these two methods
had consistent results although $\rho_{eff}^{II}$ were generally smaller than $\rho_{eff}^{I}$.
**3.1.5 Shape factor**
The shape of particles has been suggested to play an important role in their
optical properties (Zhang et al., 2008) and mixing state (China et al., 2013). Shape
factor was introduced to account for the ratio of the drag forces on a particle due to
nonspherical/irregular shape. Shape factor, which can be extracted based on the
measurement of particle density and mass has been introduced in Section 2.3.1.
We calculated the ratio of the estimated $m_p$ to the exact $m_p$ as a function of $d_m$
and χ (shown in Figure S5). For nonspheical particles (χ > 1), the estimated mass was
larger than the actual mass. We calculated the estimated mass using the exact $\rho_{eff}^{II}$
measured by the DMA-SPAMS to replace the $\rho_{eff}^{I}$ in Equation (2) as well. The
ratios of the estimated mass by this mean to the exact mass for 200, 300, 350, and 400
nm mobility selected particles were 1.4, 1.3, 1.3, and 1.2 respectively (red dots in
Figure S5). Thus, we could estimate the χ of the particle measured using the
DMA-SPAMS in the size range of 200-400 nm. Totally, the χ of 200-400 nm biomass
burning particle in this work exceeded 1.2 (~1.2-2.2). The χ decreased with the
increase of $d_m$ while the effective density showed the same trend. The more regular
shape and lower effective density of 400 nm particles compared with that of 200 nm
particles could be due to the particle chemical composition and particle voids
(discussed in Section 3.2).
**3.2 Size-resolved chemical composition**
Based on the mass spectra of single particles, the biomass burning particles were
classified into 6 particle types: 1) BB-CN: biomass burning (BB) particles with a
strong CN$^-$ ($m/z$ -26 [CN$^-$]) peak; 2) BB-EC: BB particles with strong elemental
carbon clusters (C$_n^{+/-}$); 3) BB-Nitrate: BB particles with strong nitrate ($m/z$ -46[NO$_2^-$],





-62[$NO_3^-$]) signals; 4) BB-Sulfate: BB particles with strong sulfate ($m/z$ -97[$HSO_4^-$])
signals; 5) BB-KCL: BB particles with strong potassium chlorine ($m/z$ 113[$K_2Cl^+$])
signals; and 6) BB-OC: BB particles with strong organic carbon peaks (e.g., $m/z$
27[$C_2H_3^+$], 37[$C_3H^+$], 43[$C_3H_7^+$], 51[$C_4H_3^+$], et al.). The naming of the chemical
classes is based on some of the dominant chemical species in an attempt to keep the
names short. The mass spectra for each particle type are presented in Figure S6. The
percentages of 6 particle types in different modes of aerodynamic size distribution for
200 nm and 400 nm mobility selected particles are shown in Figure 3. For 200 nm
mobility selected particles, the dominant particle types were BB-EC and BB-CN. The
percentages of particle types within the two aerodynamic modes differ slightly.
Compared with the first mode, the second mode contains more BB-CN (24.4% vs.
29.6%), more BB-KCl (1.0% vs. 4.3%) and less BB-EC (32.2% vs. 22.9%). We
supposed that the density of each particle type largely depended on the dominant
species. The exact effective density of each particle type could not be obtained
directly while the relative value compared with other particle types could be inferred
from the material density of dominant species. For example, the BB-KCl type may
have larger effective density compared with others since the dominant composition of
BB-KCl type has material density of ~1.99 g/cm$^3$ (Lide, 2008). The increased
BB-KCl type and the decrease of BB-EC (~1.0 g/cm$^3$) resulted in a larger effective
density in the second mode than the first mode.

The fractional distributions of the 6 particle types for 200 nm and 400 nm

mobility selected particles were apparently different (Figure 3). For 400 nm mobility
selected particles, the proportions of BB-Nitrate, BB-Sulfate and BB-KCl types were
larger than those of 200 nm mobility selected particles. The dominant chemical
species for BB-Nitrate and BB-Sulfate particle types could be $NH_4NO_3$ and
$(NH_4)_2SO_4$ with material density of ~1.75 g/cm$^3$ (Lide, 2008). Compared with other
types, BB-Nitrate, BB-Sulfate and BB-KCl were particle types with larger density.
However, the effective density for 400 nm mobility selected particles was smaller
than that of 200 nm. In addition to the compositional differences, particle morphology
could be another reason responsible for the observed differences in the effective
densities between these two sizes. Indeed, it has been found that the morphology like
void ratio, particle shape factor, and fractal dimension of particles all greatly affect
particle effective density (DeCarlo et al., 2004). Though the shape factor discussed in
Section 3.1 had shown that the 400 nm ($d_m$) particles had a more spherical
morphology, their lower average effective density compared to smaller particles could
be due to the voids in particles. Amorphous species such as $NH_4NO_3$ could lead to the
low effective density of particles. Thus, we supposed the lower effective density of
400 nm particles compared with 200 nm particles was caused by the large proportion
of $NH_4NO_3$ and $(NH_4)_2SO_4$ with fluffy material properties.

For 400 nm mobility selected particles, there was no big difference of particle

types between the dominant and the secondary modes. The difference between
effective densities of these two modes could be due to the different proportions of
particle types. However, the proportion of BB-KCl in the third mode at 840 nm with
effective density of 2.10 g/cm$^3$ greatly increased compared with the first two modes




(8.8%, 9.2% vs. 32.7%). The increased BB-KCl indicated that the KCl crystals were
external mixed and tended to be mixed with larger size particles which were
consistent with the DMA-APM-CPC result.
Upon heating by TD, the proportions of BB-CN and BB-KCl increased,
indicating that these types of particles were composed of less volatile species (shown
in Figure S3) (Zhai et al., 2015). At 300$^o$C, the fractions of BB-Nitrate and
BB-Sulfate decreased, consistent with the volatilization temperature ranges of
ammonium nitrate (~48-89 $^o$C) and ammonium sulfate (~178-205 $^o$C) (Johnson et al.,
2004a; Johnson et al., 2004b). The high effective density (>2.0) of biomass burning
particles at 300$^o$C could be due to the vaporization of volatile organics with small
density and the secondary species ($NH_4NO_3$ and $(NH_4)_2SO_4$ with density of ~1.75
g/cm$^3$).

### 3.3 Size-resolved optical properties
### 3.3.1 Single scattering albedo (SSA)

The single scattering albedo (SSA), was calculated using the following equation:

$$SSA\,(\lambda) = b_{scat}\,(\lambda)\,/\,[b_{abs}\,(\lambda) + b_{sca}\,(\lambda)]$$

where $b_{scat}$ is the particle light scattering coefficient, $b_{abs}$ is the light absorption
coefficient, and $\lambda$ is wavelength. The light scattering and extinction coefficients ($b_{ext}$,
$= b_{abs} + b_{sca}$) for biomass burning particles in this work were measured at 530 nm and
450 nm wavelengths using CAPSs.
The size-resolved SSAs for biomass burning particles are shown in Figure 4.
Totally, the SSAs for biomass burning particles in the mobility size range of 50-400
nm varied narrowly. However, it's worth noting that our measurement of optical
properties was based on bulk measurement by CAPSs rather than single particle like
the chemical information obtained by SPAMS. Thus, the mixing states of
size-resolved biomass burning particles which can strongly affect the balance of
radiative forcing by BC and organic matter might be offset in our measurement.
The SSA (530 nm) for 50 nm particles was the lowest (0.889±0.006) as the
percentage of strong light-absorbing black carbon for particles in this size range was
larger (shown in Figure 3, discussed in Section 3.2). For 100-400 nm biomass burning
particles, the SSAs were relatively steady (0.897±0.006 - 0.900±0.006).
The size-resolved SSAs at 450 nm ($\lambda$) for biomass burning particles were
generally lower than those at 530 nm ($\lambda$). Previous studies have shown that biomass
burning was an important source of brown carbon (BrC) which is light-absorbing in
the UV-vis range (Lack and Cappa, 2010). For 50 nm ($d_m$) particles, the SSA ($\lambda$=450
nm) was also the lowest, due to the dominance of the strong light-absorbing BC in
these particles. However, unlike the trend of size-resolved SSAs ($\lambda$=530 nm), the SSA
($\lambda$=450 nm) of 100-400 nm particles decreased as the size increased. It has been
shown that brown carbon arising from biomass burning is primarily composed of
extremely low volatility organic compounds (Saleh et al., 2014). The CN$^-$ in biomass
burning particles is representative for some extremely low-volatile
nitrogen-containing organics (Zhai et al., 2015). As shown in Figure 3, compared with
400 nm particles, the proportion of organic matter (BB-CN, BB-OC) was larger for
200 nm particles. The nitrogen-containing species might indicate the existence of





light-absorbing organics. The lower SSA (λ=450 nm) for 200 nm particles might
indicate higher proportion of BrC. We assumed the lower SSA (λ=450 nm) for 100
nm performed in a similar way with larger proportion of BrC.

**3.3.2 Ångström absorption exponent (AAE)**

To investigate the wavelength dependence of the absorption coefficients, we
determined the Ångström absorption exponent (AAE) based on absorption
measurements at two different wavelengths (λ1 & λ2) using the following equation:
$$\text{AAE} (\lambda_1 / \lambda_2) = - \ln[ b_{abs} (\lambda_1) / b_{abs} (\lambda_2)] / \ln(\lambda_1 / \lambda_2)$$
The AAE in this work was calculated from the light absorption coefficients at
wavelengths of 450 nm and 530 nm measured by the CAPSs. The uncertainties in the
calculated AAE values can be caused by the uncertainties in the calibration factors of
CAPSs. The size-resolved AAEs for biomass burning particles are shown in Figure 4.
Black carbon is highly absorbing in the visible spectrum with little variation with
wavelength and shows an AAE of ~1.0 (Bergstrom et al., 2002). As brown carbon
species absorb light in the UV-vis range, BrC-containing particles usually exhibit an
AAE above 1 (Martinsson et al., 2015). Lack and Cappa (2010) used modeling to
calculate AAE values and suggested that particles with AAE exceeding 1.6 should be
classified as BrC. In our study, the AAE values of particles in the size range of 50-400
nm were higher than 1.6, indicating that they were BrC containing particles from
biomass burning. Among all sizes, the AAE of 50 nm biomass burning particles was
the lowest (~5.8) while that of 100 nm particles was the highest (~6.3). The main
light-absorbing functional groups in the UV-vis range is conjugated double bond
(Laskin et al., 2015). BB-CN and BB-OC particle types identified by mass spectra in
our work tended to contain more large molecules of BrC with light-absorbing
functional groups. We noticed that the proportion of BB-OC type species was larger in
200 nm particles (Figure 3) and with higher AAE value, compared with 400 nm
particles. Thus, we suppose the highest AAE value observed for 100 nm particles
might be the result of the highest BrC proportion.
The SSA and AAE values of total biomass burning particles are shown in Table
S2. The decrease of SSA values upon heating was due to the vaporization of
secondary inorganic species like $NH_4NO_3$ and less absorbing organics. The AAE
values for all particles at 150 °C and 300 °C were ~19% and ~64%, respectively,
lower than those at room temperature (20 °C). The large decrease of AAE at 300 °C
could be due to the vaporization of light-absorbing organics in the temperature range
of 150-300 °C. However, the AAE value at 300 °C was still above the standard of BrC
(1.6), indicating the presence of extremely low-volatile light-absorbing organics in
biomass burning particles.

**3.3.3 Absorption enhancement ($E_{abs}$)**

The impact of other particle components on BC absorption, either internally or
externally mixed of BC with organic aerosol and inorganic salts, has drawn
significant attention. The light absorption by an absorbing core can be enhanced when
coated with a purely scattering shell which acts as a lens. Absorption enhancement
has been observed in laboratory for BC particles coated with various materials
(Schnaiter et al., 2005; Zhang et al., 2008), and in field observation (Schwarz et al.,



2008; Spackman et al., 2010). In this study, we measured the absorption enhancement
of freshly emitted straw combustion particles.
The light absorption enhancement ($E_{abs}$) due to coating was estimated by the
ratio of $b_{abs}(\lambda)$ for particles that did and did not pass through the TD:

$$E_{abs}(\lambda, T) = b_{abs}(\lambda, T_0)/ b_{abs}(\lambda, T)$$

where $T$ is the TD temperature (150 or 300 $^{o}$C), $T_0$ is the room temperature (20 $^{o}$C in
this work).
As heating by the TD, the particles might shrink to smaller sizes. A tandem
DMAs (TDMA) was utilized to detect the size change of particles. Here, we used the
ratio of the particle diameter after heating ($d_{m2}$) to the diameter before heating ($d_{m1}$) as
the shrink factor ($d_{m2}/d_{m1}$) of particles (shown in Figure S7). The absorption
coefficient of particles was calibrated by the shrink factor and transport efficiency as
mentioned in Section 2.5.
The size-resolved $E_{abs}$ observed at wavelengths of 530 nm and 450 nm are shown
in Figure 5. Though freshly emitted, absorption enhancements ($E_{abs}$) of biomass
burning particles in the size range of 50-400 nm were observed ($E_{abs}>1$). Totally, the
$E_{abs}$ increased with increasing particle diameters with the largest $E_{abs}$ ($\lambda=530$ nm) of
1.197$\pm$0.082 and the $E_{abs}$ ($\lambda=450$ nm) of 1.460$\pm$0.101 for 400 nm particles. This
could be explained by the thicker coating (both primary and secondary organic and
inorganic species) for larger particles. The $E_{abs}$ ($\lambda=450$ nm) were overall larger than
the $E_{abs}$ ($\lambda=530$ nm). You et al. (2016) reported that the $E_{abs}$ of BC internally mixed
with humic acid (HA/BC) ranged from 2 to 3 and was strongly wavelength dependent.
Removal of the HA absorption contribution revealed the independence of wavelength.
Thus, the larger $E_{abs}$ ($\lambda=450$ nm) in this work could be due to the absorption of
light-absorbing organics.

**4 Conclusions**
As a major primary source of aerosols, biomass burning emissions significantly
impact the global radiative budget. It is important to understand the physical and
chemical properties of biomass burning particles, as well as their links to optical
properties. In this work, rice straw was combusted as a representative material of
biomass burning in Southern China. A series of comprehensive methods was used to
detect the size-resolved chemical composition, effective density, and optical
properties of the particles emitted from the burns.
Two methods were utilized to measure the effective density of the biomass
burning particles. The DMA-APM-CPC system, which has been widely used in
chamber and field work, offered size-resolved information on the particle effective
density. The DMA-SPAMS system provided physical property and chemical
composition at single-particle level. The 50 nm ($d_m$) biomass burning particles had the
lowest effective density of 1.15$\pm$0.23 g/cm$^3$, which was due to the large portion of
aggregate black carbon. The apparent single-peak density distribution of 50 nm
particles was due to the combination of two modes (BC and organic matter,
respectively). The independent modes at 0.9-1.1 g/cm$^3$ shown in the density
distribution of 100 nm and 200 nm particles and ~1.92 g/cm$^3$ mode shown in that of





400 nm particles indicated that BC and crystalline species such as KCl in fresh biomass burning particles tended to be externally mixed with organic carbon. With heating by TD, the separation of the effective density distribution modes testified the presence of BC, potassium salts and less volatile OC in the biomass burning particles.

The effective density measured by DMA-SPAMS system was consistent with the result by DMA-APM-CPC method. The dominant modes in the effective density distributions of 200 nm and 400 nm mobility selected particles were 1.40 g/cm$^3$ and 1.35 g/cm$^3$, respectively. The crystalline KCl with an effective density of 2.10 g/cm$^3$ (with BB-KCl type accounting for 32.7%) was observed in the density distribution for 400 nm particles measured by DMA-SPAMS. The proportions of BB-Nitrate, BB-Sulfate, and BB-KCl types in 400 nm mobility selected particles were larger than those in 200 nm mobility selected particles. Compared with 200 nm particles, 400 nm particles showed more spherical morphology but lower effective density, which could be due to the larger proportion of amorphous $NH_4NO_3$.

The size-resolved extinction and scattering coefficients were measured by CAPSs at wavelengths of 450 nm and 530 nm. The SSA ($\lambda$=530 nm) for 50 nm particles was the lowest (0.889±0.006) because of the presence of a larger percentage of the strongly light-absorbing black carbon particles in this size mode. The size-resolved SSAs ($\lambda$=450 nm) for biomass burning particles were generally lower than the SSAs ($\lambda$=530 nm). The AAE values in the size range of 50-400 nm particles were all above 1.6, the acceptable standard of brown carbon. The AAE value was the lowest for 50 nm particles (~5.8) while was the highest for 100 nm particles (~6.3). Compared with 400 nm particles, the proportions of BB-OC and BB-CN, the extremely low-volatile organic compounds, were larger for 200 nm particles which might indicate a higher possibility for the existence of light-absorbing organics. The $E_{abs}$ was observed in freshly emitted biomass burning particles. The $E_{abs}$ increased with larger diameter due to increasing coating thickness. The wavelength-dependent $E_{abs}$ of particles were likely due to the absorption of light-absorbing organics. Our work emphasizes on the complex mixing states of aerosols from primary source. Further research on how particle morphology affects the optical properties of biomass burning particles is needed.

**Acknowledgements**

This work was supported by the National Natural Science Foundation of China (91544224, 21507010), the Ministry of Science & Technology of China (2012YQ220113-4), the Science & Technology Commission of Shanghai Municipality (14DZ1202900), and the Changjiang Scholars program of the Chinese Ministry of Education.

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

Contributions of brown carbon and lensing effect, J. Geophys. Res.-Atmos., 119,
12721-12739, doi: 10.1002/2014jd021744, 2014.
Onasch, T. B., Massoli, P., Kebabian, P. L., Hills, F. B., Bacon, F. W., and Freedman,
A.: Single scattering albedo monitor for airborne particulates, Aerosol Sci. Technol.,
49, 267-279, doi: 10.1080/02786826.2015.1022248, 2015.
Park, R., Jacob, D., Kumar, N., and Yantosca, R.: Regional visibility statistics in the
United States: Natural and transboundary pollution influences, and implications for





the Regional Haze Rule, Atmos. Environ., 40, 5405-5423, doi:
10.1016/j.atmosenv.2006.04.059, 2006.
Petters, M. D., Carrico, C. M., Kreidenweis, S. M., Prenni, A. J., DeMott, P. J., Collett,
J. L., and Moosmuller, H.: Cloud condensation nucleation activity of biomass
burning aerosol, J. Geophys. Res.-Atmos., 114, 16, doi: 10.1029/2009jd012353,
2009.
Philippin, S., Wiedensohler, A., and Stratmann, F.: Measurements of non-volatile
fractions of pollution aerosols with an eight-tube volatility tandem differential
mobility analyzer (VTDMA-8), J. Aerosol Sci., 35, 185-203, doi:
10.1016/j.jaerosci.2003.07.004, 2004.
Pitz, M., Schmid, O., Heinrich, J., Birmili, W., Maguhn, J., Zimmermann, R.,
Wichmann, H. E., Peters, A., and Cyrys, J.: Seasonal and diurnal variation of
PM2.5 apparent particle density in urban air in Augsburg, Germany, Environ. Sci.
Technol., 42, 5087-5093, doi: 10.1021/es7028735, 2008.
Saleh, R., Robinson, E. S., Tkacik, D. S., Ahern, A. T., Liu, S., Aiken, A. C., Sullivan,
R. C., Presto, A. A., Dubey, M. K., Yokelson, R. J., Donahue, N. M., and Robinson,
A. L.: Brownness of organics in aerosols from biomass burning linked to their
black carbon content, Nat. Geosci., 7, 647-650, doi: 10.1038/ngeo2220, 2014.
Schnaiter, M., Linke, C., Mohler, O., Naumann, K. H., Saathoff, H., Wagner, R.,
Schurath, U., and Wehner, B.: Absorption amplification of black carbon internally
mixed with secondary organic aerosol, J. Geophys. Res.-Atmos., 110, doi:
10.1029/2005jd006046, 2005.
Schwarz, J., Spackman, J., Fahey, D., Gao, R., Lohmann, U., Stier, P., Watts, L.,
Thomson, D., Lack, D., and Pfister, L.: Coatings and their enhancement of black
carbon light absorption in the tropical atmosphere, J. Geophys. Res.-Atmos., 113,
doi: 10.1029/2007JD009042, 2008.
Song, X. H., Hopke, P. K., Fergenson, D. P., and Prather, K. A.: Classification of
single particles analyzed by ATOFMS using an artificial neural network, ART-2A,
Anal. Chem., 71, 860-865, doi: 10.1021/ac9809682, 1999.
Spackman, J. R., Gao, R. S., Neff, W. D., Schwarz, J. P., Watts, L. A., Fahey, D. W.,
Holloway, J. S., Ryerson, T. B., Peischl, J., and Brock, C. A.: Aircraft observations
of enhancement and depletion of black carbon mass in the springtime Arctic, Atmos.
Chem. Phys., 10, 9667-9680, doi: 10.5194/acp-10-9667-2010, 2010.
Spencer, M. T., Shields, L. G., and Prather, K. A.: Simultaneous measurement of the
effective density and chemical composition of ambient aerosol particles, Environ.
Sci. Technol., 41, 1303-1309, doi: 10.1021/es061425+, 2007.
Tajima, N., Fukushima, N., Ehara, K., and Sakurai, H.: Mass range and optimized
operation of the aerosol particle mass analyzer, Aerosol Sci. Technol., 45, 196-214,
doi: 10.1080/02786826.2010.530625, 2011.
Tang, I. N., and Munkelwitz, H. R.: Water activities, densities, and refractive indices
of aqueous sulfates and sodium nitrate droplets of atmospheric importance, J.
Geophys. Res.-Atmos., 99, 18801-18808, doi: 10.1029/94jd01345, 1994.
Turpin, B. J., and Lim, H.-J.: Species contributions to PM2.5 mass concentrations:
revisiting common assumptions for estimating organic mass, Aerosol Sci. Technol.,



35, 602-610, doi: 10.1080/02786820119445, 2001.
Wentzel, M., Gorzawski, H., Naumann, K. H., Saathoff, H., and Weinbruch, S.:
Transmission electron microscopical and aerosol dynamical characterization of soot
aerosols, J. Aerosol Sci., 34, 1347-1370, doi: 10.1016/s0021-8502(03)00360-4,
2003.

You, R., Radney, J. G., Zachariah, M. R., and Zangmeister, C. D.: Measured
Wavelength-Dependent Absorption Enhancement of Internally Mixed Black
Carbon with Absorbing and Nonabsorbing Materials, Environ. Sci. Technol., doi:
10.1021/acs.est.6b01473, 2016.

Zhai, J., Wang, X., Li, J., Xu, T., Chen, H., Yang, X., and Chen, J.: Thermal
desorption single particle mass spectrometry of ambient aerosol in Shanghai,
Atmos. Environ., 123, 407-414, doi: 10.1016/j.atmosenv.2015.09.001, 2015.

Zhang, R., Khalizov, A. F., Pagels, J., Zhang, D., Xue, H., and McMurry, P. H.:
Variability in morphology, hygroscopicity, and optical properties of soot aerosols
during atmospheric processing, P. Natl. Acad. Sci. USA, 105, 10291-10296, doi:
10.1073/pnas.0804860105, 2008.

Zhang, Y., Zhang, Q., Cheng, Y., Su, H., Kecorius, S., Wang, Z., Wu, Z., Hu, M., Zhu,
830       T., Wiedensohler, A., and He, K.: Measuring the morphology and density of
internally mixed black carbon with SP2 and VTDMA: new insight into the
absorption enhancement of black carbon in the atmosphere, Atmos. Meas. Tech., 9,
1833-1843, doi: 10.5194/amt-9-1833-2016, 2016






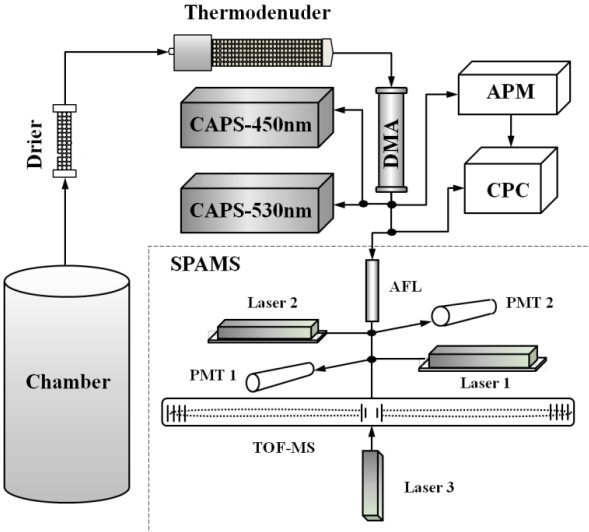


Figure 1. Schematic of the instrumental setup. The CAPS, DMA, CPC, APM and
SPAMS represent Cavity Attenuated Phase Shift spectroscopy, Differential Mobility
Analyzer, Condensation Particle Counter, Aerosol Particle Mass analyzer and Single
Particle Aerosol Mass Spectrometer, respectively.





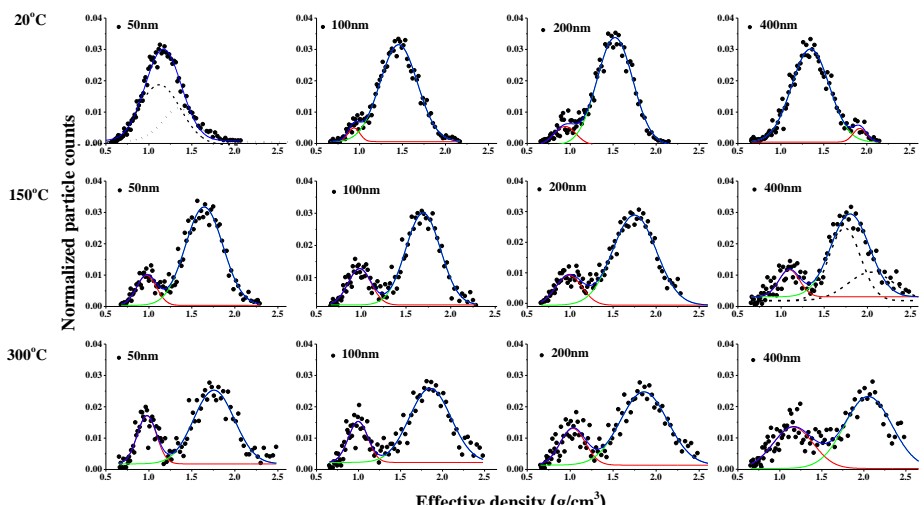


Figure 2. Average density distributions of 50, 100, 200, and 400 nm particles selected
by DMA at 20 °C (room temperature), 150 °C, and 300 °C. Gaussian model was
applied in fitting each density scan (red and green lines). Black dashes were the
assumption Gaussian models application.





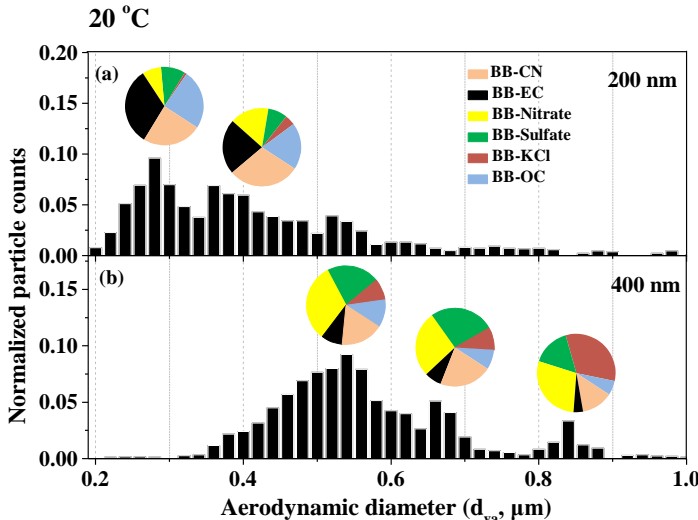


Figure 3. Vacuum aerodynamic size distributions detected by the SPAMS of 200 nm
(a) and 400 nm (b) electrical mobility size-selected biomass burning particles and pie
charts for the particle types in different aerodynamic modes at 20 $^{o}$C (room
temperature).





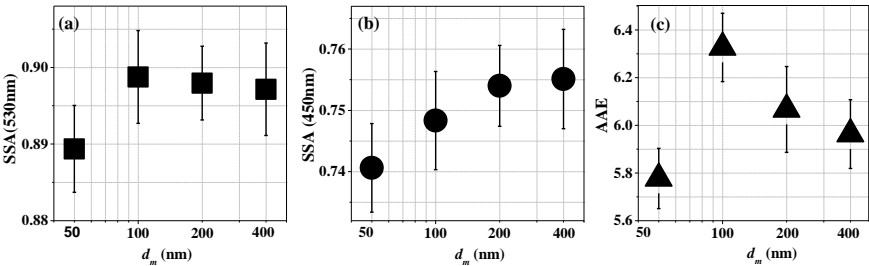


Figure 4. (a)-(b): Size-resolved single scattering albedo (SSA) at wavelengths of 530
nm and 450 nm. (c): Ångström absorption exponent (AAE) of biomass burning
particles at room temperature (20°C).



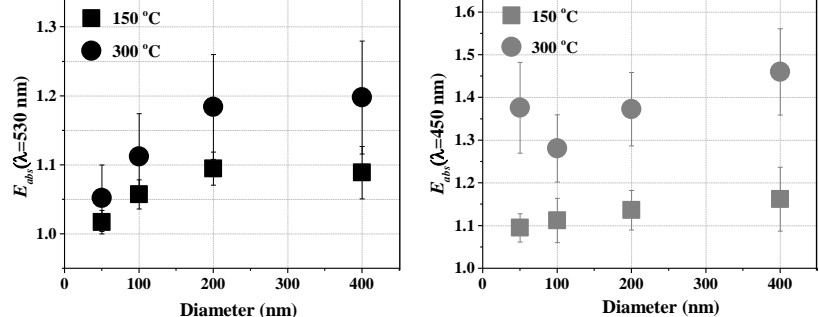


Figure 5. The size-resolved absorption enhancement ($E_{abs}$) at wavelengths of 450 nm
and 530 nm.