# Peer review of "Size-resolved chemical composition, effective density, and optical"

_Atmospheric Chemistry and Physics, 2016_

## Referee Comment (RC1) · Anonymous Referee #1 · 5 Jan 2017

This article presents measurements of density, optical properties and chemical composition of biomass particles produced from the combustion of rice straw. In general, I think this study employs an impressive suite of instrumentation to characterize the chemical, optical and physical properties of biomass burn particles. The article does lack a main point and there are some deficiencies in presentation. Moreover, some of the conclusions do not follow from the data as mentioned in the "Detailed Comments" that follow. The main technical problems are: 1) the inference of ammonium nitrate and sulfate as major components of the aerosol without data to back up that claim, 2) the use of thermodenuding and bulk optical property measurement to infer things about coatings; I think there are some major assumptions that need to be addressed before

confidently reaching this conclusion, 3) the methods utilized to calculate effective densities are not adequately described, 4) there is a body of literature from the FLAME experiments that I think could be used to better interpret these results, 5) the composition measurements can be better analyzed and presented, 6) the article needs to be better organized around a main point, and 7) I think the internal mixing of organics influence the effective density and this does not come through very well in the manuscript while I think it can be inferred with the available data. In many places, the tables and figures can be described better and I have done my best to point that out. I think the article can be better proofread for grammar as well. In light of this critique, I recommend that the article undergo a major rewrite before it is acceptable for publication in ACP. The article will be a great addition to the literature after a significant transformation. Detailed Comments: L 24: "relative" should be "relatively" L 54: awkward sentence L 59: Surely the authors can provide more current refrences on BC – such as Tami Bond's extensive article published in JGR called "bounding black carbon". L 60: awkward sentence L 70: suggest changing "low-visible" to "short wavelength visible" L 89: The Tang an Munkelwitz article probably not the best reference here. L 105: This seems to be incorrect – bulk measurements cannot distinguish between particles in a population. L 114: grammar L 116: morphology of BB particles has been extensively documented via microscopy measurements. I think the introduction should include some of these. Most notably, Hopkins et al analyzed the optical and morphological properties of rice straw from the Flame experiment: Hopkins, R. J., K. Lewis, Y. Desyaterik, Z. Wang, A. V. Tivanski, W. P. Arnott, A. Laskin, and M. K. Gilles (2007), Correlations between optical, chemical and physical properties of biomass burn aerosols, Geophys. Res. Lett., 34, L18806, doi:10.1029/2007GL030502. L 119: Grammar L 234: The procedures outlined in this section are not clear. Why are densities assumed to be unity? How will these assumptions be cancelled out later? What is being calculated and what procedures (e.g. iterative solutions...etc.) are being used? L 241: This title is too general to be of any use. L 249: change to Kr-85 (use a dash) L 297 and Table S1: Xc1 and Xc2 need to be defined - I assume these are some indication the peak positions. Without knowing this it is difficult to make sense of the table. How did the peak positions change for the different fits? What do the numbers in the Table S1 indicate? Area? Height? L 312-317: What temperature is being referred to? I think the density profile for the room temperature particles is NOT bimodal. The data for (what is presumably) the second mode is noisy. Furthermore, figure S1 seems to have some fit parameters that need to be more fully described. The bimodal gaussian fits should be used in the discussion. To state that the two components are externally mixed after passing through the thermodenuder is misleading. L 327: Whos to say that internally mixed OC and potassium salts do not contribute to modes with lower densities? In fact, I think data shown in Figure 3 and S3 support the internal mixing of potassium and organics. The mass spectra shown in Figure S6 also support this. Its not clear why they did not look at the chemical changes as a function of temperature. L 336: Organic matter can be secondary, so the parenthetic statement is inaccurate. L 349: What is "deviation range"? It would be useful to mark out suspected doubly charged modes on the figure.

L 356: "bi-model" should be "bimodal". L 392: Why is one method for obtaining "effective density" consistently lower than the other? Can other information be extracted from this? I dont understand the value of reporting results from the two methods. L 434: I highly doubt the material density of the BB-KCl type is in the CRC handbook – in fact the CRC is referenced repeatedly in weird places. L 436: Which is the "first" and which is the "second" mode? L 442: I think this should be better referenced. What data suggests ammonium nitrate and sulfate are the dominant composition from rice straw burning? I dont think the measured density is a reliable way to infer chemical composition due to the fact that the material density may be a weighted average of organic and inorganic species. If ammonium salts are so prevalent, one would expect $m/z = 18$ to be fairly prominent - this is not mentioned in the manuscript. L 452: Is ammonium nitrate really expected to be amorphous? This needs to be proven (with references), otherwise it is very speculative. L 457: these modes are hardly discernable - especially the middle mode in figure 3b. L 464-472: I think the differences in SPMS data at the different temperatures is very telling. The drastic decrease in the OC cluster at high

temperatures can explain why the effective density increases: these particles loose low density organics and become more dense. I think this point is missing from the discussion of the densities. The loss of organics and other low volatility secondary species is not suprising and can be shown better thorugh the use of difference spectra. Furthermore, I do not see why Figures 3 and S3 are separate. I think it would be much more impactful to show these figures together. L 487: what is offset here? Forcing? This is unclear. L 560: This sentence is unclear. Furthermore, the procedures referred to in section 2.5 are not really described in that section, so I suggest that the authors give this some detail. Another more general comment about using thermodenuders to estimate absorption enhancement: how might the TD cause side reactions to affect optical properties? L 567: The above definition (L 553) of absorption enhancement is really a bulk definition. Without knowledge of the exact mixing state of the population, I think it is difficult to attribute "absorption enhancement" to mixing state effects, no? L 589: "nonspherical" or "fractal" may be a better term to use in place of "aggregate". L 606: again I think lower densities may be caused by mixing with organics – as described above. L 613: "acceptable standard" does not seem right here. What is "typical"? L 616: change "volatile" to "volatility" L 619: How can the authors attribute absorption enhancement to coating thickness using bulk measurements? Here the conclusions do not really follow from the data.

---

## Referee Comment (RC2) · Anonymous Referee #2 · 6 Jan 2017

Zhai and coauthors describe a series of experiments used to investigate the chemical, physical and optical properties of particles generated through the combustion of rice straw, chosen to represent local crop residues in Southern China. The motivation is to provide data that will help modelers to better assess regional and global radiative climate impacts of particles from this source. A combination of instrumentation is applied to calculate effective densities in two ways, employing a scanning mobility particle sizer, single particle mass spectrometer and an aerosol particle mass analyzer. Extinction, scattering and calculated absorption coefficients are also derived from a cavity attenuated phase shift spectrometer. These properties are assessed for untreated and thermally denuded rice straw combustion particles. While the experiments

are worthwhile, there are some issues regarding the approaches taken, and the results need to be framed in the context of existing work in a more rigorous way. Suggested revisions are below:

Major comments:

Why is the rice straw dried at 100 °C prior to use? Presumably local farmers do not do this and simply burn the residue with its natural water content. Wouldn't this change the burn conditions, smouldering etc? Would it be more representative to burn untreated fuel?

Why is a relatively low laser desorption ionization energy (0.6 mJ per pulse) selected here? Routinely single particle mass spectrometers of this design employ energies on the order of 1 mJ per pulse. How were the ART-2a cluster parameters chosen- are they simply selected from previous work? If so cite the source and reason for selection.

Is a flow rate of 0.6 L min-1 within the manufacturer's operating range for the thermal denuder?

The transmission efficiency of the thermal denuder was tested using NaCl particles. Why not simply examine the transmission efficiency of size selected biomass combustion particles at room temperature to validate agreement with the NaCl transmission efficiencies at room temperature. Also, for the absorption enhancement calculation, the transmission efficiency needs to be taken into account. The formula on line 553 simply relates babs before and after the denuder. Any losses of BC through diffusion and impaction in the denuder will be incorrectly assigned as absorption reduction through a removal of coating materials (absorption enhancement effect). Furthermore, why is the "shrink factor" used to account for volatilized coating materials? Firstly, the shrink factors shown in Fig S7 demonstrate a wide range of values, with multiple modes, and using a single value does not seem appropriate. In any case accounting for shrinking should not be necessary. Shrinking of the coatings is desirable for this test. The idea is that the bulk BC mass should be conserved through the denuder (applying the

correct transmission efficiency factor should result in BC mass entering the denuder being roughly equivalent to "transmission corrected" BC mass exiting the denuder), but that the coating materials should evaporate. Any reduction in absorption can thus be assigned to either reduced optical "lensing" due to the reduction of coating sizes or evaporation of any non-BC absorbing material. Applying the transmission efficiency correction and removing the shrink factor should be tested.

The x-axis of Figure 3 should show include effective density also. It would also be very useful to show an effective density distribution plot for each of the different particle classes. This will allow the authors to provide better evidence to support their postulations regarding potential composition-density relationships throughout.

Overall, there needs to be much better discussion of the findings here in the context of similar work previously performed by other groups. In terms of single particle mass spectrometry analyses of biomass burning particles (and relating biomass particle composition to optical and physical properties) there is a host of relevant studies that should be discussed. See (Silva et al., 1999; Zauscher et al., 2013; Bi et al., 2011; Guazzotti et al., 2003; Schwarz et al., 2008; Moffet and Prather, 2009; Moffet et al., 2008; Pagels et al., 2013). In terms of efforts examining relationships between biomass burning chemical composition measured through mass spectrometry and optical absorption enhancements see (McMeeking et al., 2014; Healy et al., 2015) and other ambient combustion particle absorption enhancements (Cappa et al., 2012; Liu et al., 2015) among others. One point which is posed as a new conclusion (evidence for external mixing of K and organic aerosol in biomass burning particles using single particle mass spectrometry) has been recently reported by others (Lee et al., 2016).

Minor comments:

Some SI figures should be in the main manuscript (S3 and S6)

Figure S4: the legend does not specify the method used for effective density calculation

Section 3.2: Do all of the particle types need the prefix BB? All particles discussed are from biomass. Also, the order of the particle types does not match Fig S6

Line 392: Test for statistically significant differences

Line 386: Simply remove the intermediate size from the plot if it is not represented in the other plot

Line 453: Is there a way to demonstrate this? See suggestion of effective density plots for each particle class earlier

Line 487: This is not an offset, simply a bulk measurement.

References

Bi, X., Zhang, G., Li, L., Wang, X., Li, M., Sheng, G., Fu, J., Zhou, Z., 2011. Mixing state of biomass burning particles by single particle aerosol mass spectrometer in the urban area of PRD, China. Atmospheric Environment 45, 3447-3453.

Cappa, C.D., Onasch, T.B., Massoli, P., Worsnop, D.R., Bates, T.S., Cross, E.S., Davidovits, P., Hakala, J., Hayden, K.L., Jobson, B.T., Kolesar, K.R., Lack, D.A., Lerner, B.M., Li, S.-M., Mellon, D., Nuaaman, I., Olfert, J.S., Petäjä, T., Quinn, P.K., Song, C., Subramanian, R., Williams, E.J., Zaveri, R.A., 2012. Radiative Absorption Enhancements Due to the Mixing State of Atmospheric Black Carbon. Science 337, 1078-1081.

Guazzotti, S.A., Suess, D.T., Coffee, K.R., Quinn, P.K., Bates, T.S., Wisthaler, A., Hansel, A., Ball, W.P., Dickerson, R.R., Neusüß, C., Crutzen, P.J., Prather, K.A., 2003. Characterization of carbonaceous aerosols outflow from India and Arabia: Biomass/biofuel burning and fossil fuel combustion. J. Geophys. Res. 108.

Healy, R.M., Wang, J.M., Jeong, C.H., Lee, A.K.Y., Willis, M.D., Jaroudi, E., Zimmerman, N., Hilker, N., Murphy, M., Eckhardt, S., Stohl, A., Abbatt, J.P.D., Wenger, J.C., Evans, G.J., 2015. Light-absorbing properties of ambient black carbon and brown carbon from fossil fuel and biomass burning sources. Journal of Geophysical Research:

Atmospheres 120, 6619-6633.

Lee, A.K.Y., Willis, M.D., Healy, R.M., Wang, J.M., Jeong, C.H., Wenger, J.C., Evans, G.J., Abbatt, J.P.D., 2016. Single-particle characterization of biomass burning organic aerosol (BBOA): evidence for non-uniform mixing of high molecular weight organics and potassium. Atmos. Chem. Phys. 16, 5561-5572.

Liu, S., Aiken, A.C., Gorkowski, K., Dubey, M.K., Cappa, C.D., Williams, L.R., Herndon, S.C., Massoli, P., Fortner, E.C., Chhabra, P.S., Brooks, W.A., Onasch, T.B., Jayne, J.T., Worsnop, D.R., China, S., Sharma, N., Mazzoleni, C., Xu, L., Ng, N.L., Liu, D., Allan, J.D., Lee, J.D., Fleming, Z.L., Mohr, C., Zotter, P., Szidat, S., Prevot, A.S.H., 2015. Enhanced light absorption by mixed source black and brown carbon particles in UK winter. Nat Commun 6.

McMeeking, G.R., Fortner, E., Onasch, T.B., Taylor, J.W., Flynn, M., Coe, H., Kreidenweis, S.M., 2014. Impacts of nonrefractory material on light absorption by aerosols emitted from biomass burning. Journal of Geophysical Research: Atmospheres 119, 2014JD021750.

Moffet, R.C., Prather, K.A., 2009. In-situ measurements of the mixing state and optical properties of soot with implications for radiative forcing estimates. Proceedings of the National Academy of Sciences 106, 11872-11877.

Moffet, R.C., Qin, X., Rebotier, T., Furutani, H., Prather, K.A., 2008. Chemically segregated optical and microphysical properties of ambient aerosols measured in a single-particle mass spectrometer. J. Geophys. Res. 113, D12213.

Pagels, J., Dutcher, D.D., Stolzenburg, M.R., McMurry, P.H., Gälli, M.E., Gross, D.S., 2013. Fine-particle emissions from solid biofuel combustion studied with single-particle mass spectrometry: Identification of markers for organics, soot, and ash components. Journal of Geophysical Research: Atmospheres.

Schwarz, J.P., Gao, R.S., Spackman, J.R., Watts, L.A., Thomson, D.S., Fahey, D.W.,
* * *
Interactive
comment

Ryerson, T.B., Peischl, J., Holloway, J.S., Trainer, M., Frost, G.J., Baynard, T., Lack, D.A., de Gouw, J.A., Warneke, C., Del Negro, L.A., 2008. Measurement of the mixing state, mass, and optical size of individual black carbon particles in urban and biomass burning emissions. Geophys. Res. Lett. 35, L13810.

Silva, P.J., Liu, D.Y., Noble, C.A., Prather, K.A., 1999. Size and chemical characterization of individual particles resulting from biomass burning of local Southern California species. Environmental Science & Technology 33, 3068-3076.

Zauscher, M.D., Wang, Y., Moore, M.J.K., Gaston, C.J., Prather, K.A., 2013. Air Quality Impact and Physicochemical Aging of Biomass Burning Aerosols during the 2007 San Diego Wildfires. Environmental Science & Technology.

---

## Referee Comment (RC3) · Anonymous Referee #3 · 28 Jan 2017

This manuscript reports the size-resolved chemical composition, effective density and optical properties of rice straw burning particles using different online instruments including DMA-APM-CPC, single particle aerosol mass spectrometer, and cavity attenuated phase shift (CAPS) spectroscopy. First of all, the focus of this study is unclear and hence the significance and atmospheric implication should be explicitly highlighted in the abstract, introduction and conclusion. Secondly, the results should be discussed in more detail. In particular, the observations regarding particle effective densities and single particle compositions should be better integrated in this manuscript in order to provide a more complete picture on the relationship of particle mixing state, morphology and effective density. Lastly, there is an issue about the data quality in

Section 3.3.3 (See specific comments) that should be addressed by the authors. The manuscript should be proofread carefully before submission. Overall, I don't recommend this manuscript to be published in Atmospheric Chemistry and Physics in the current format. Below is the specific comments.

Specific comments:

1. Absorption enhancement results (Section 3.3.3 and Figure 5): Figure 1 demonstrated that aerosol particles were pre-treated by a thermodenuder before generating monodisperse particles using a DMA. The size-selected particles were then characterized by a few real-time instruments. Using such experimental setup, the size-resolved absorption enhancement factors reported in Figure 5 are actually not meaningful. The primary reason is that those enhancement factors were not determined by comparing particles with the same original dried-particle diameter. The whole particle size distribution should shift towards the lower particle size after thermal treatment (i.e. removal of coating materials). The original size of heated particles should be greater than that of particles without heating as shown in Figure S7. It is unclear how the shrink factor and transmission efficiency of particles presented in Section 3.3.3 can resolve this fundamental problem. Detail clarification is required to keep the related discussion in the manuscript.

Furthermore, this manuscript emphasizes a few times that aerosol coating can act as a lens to enhance light absorption of aerosol particles in general. Nevertheless, the major conclusion of this work regarding light absorption enhancement (relative to pure BC) is due to the presence of atmospheric brown carbon from biomass burning emissions and without much discussion on the lensing effect. It is recommended to change the tone/wording in the text to avoid any potential confusion to the readers, especially for those are not familiar with this research topic.

2. Figure 2 and Section 3.1.3: Similar to the comment #1, it is essential to highlight in the manuscript that the individual column in Figure 2 is not presenting results obtained

from particles with the same original dried-particle diameter. The current writing is somewhat misleading. However, the findings observed from this figure is still useful even though direct comparison of plots displayed in the same column is inappropriate.

3. To better organize the discussion of particle effective density, it is recommended to combine Section 3.1.2 and 3.1.4 in the manuscript.

4. Effective density and chemical composition (Section 3.1.3): 1) Lines 358-360: Without quantitative chemical characterization in a single particle basis, it is hard to prove if some BC was externally mixed with other components based on the effective density measurement alone. BC particles with highly fractal structure and thin organic coating can lead to the similar observations, depending on the uncertainty of effective density measurement. 2) Lines 365-378: According to Figure 2, peaks are always observed at around 1.7-1.8 g/cm3 for the heated particles with diameter less than 400 nm. Please comment whether this observation is due to the presence of extremely low volatility organic aerosol materials generated. Furthermore, what are the vaporization temperature of KCl and other potassium salts.

5. Section 3.2 and Figure 3: The authors may over-interpret their observations. The average chemical compositions are not sufficient enough (i.e. they are too similar) to explain the effective densities of the two particle modes presented in Figure 3a (200 nm). Similarly, the first two particle modes in Figure 3b (400 nm) have the very similar average chemical compositions. What are the particle number distributions of each cluster (lines 430-432)? Their particle number distributions should be able to extract from the results of cluster analysis. Estimation of effective densities of some clusters is possible with such additional information. Furthermore, it is unclear how to separate the particle modes in Figure 3 for constructing the pie chart. Please provide sufficient detail.

6. Section 3.3.2, Lines 533-541: What is the particle size for data reported in Table S2? Significant drop of AAE for the thermodenuded particles are observed. Please
comment on the significance and relative contributions of extremely low volatility brown carbon to the total light absorption properties of biomass burning aerosol observed in this study and compare their results with existing literature.

Minor and technical comments:

- Please replace "bi-model" by "bi-modal" throughout the entire manuscript.

- Figure S4: The legend displays the same for both types of effective density. Please correct.

- Please use "low" and "high" to describe density.

- Line 198: Should it be "…increased as the size increased…"?

---

## Author Comment (AC1) · 14 Mar 2017

**Response to Review 1**

We sincerely thank the reviewer for the valuable comments and suggestions. Below we list our point-by-point replies to the comments and the descriptions of the changes we made in the revised manuscript.

This article presents measurements of density, optical properties and chemical composition of biomass particles produced from the combustion of rice straw. In general, I think this study employs an impressive suite of instrumentation to characterize the chemical, optical and physical properties of biomass burn particles. The article does lack a main point and there are some deficiencies in presentation. Moreover, some of the conclusions do not follow from the data as mentioned in the "Detailed Comments" that follow. The main technical problems are:

1) the inference of ammonium nitrate and sulfate as major components of the aerosol without data to back up that claim,

   **Response:** Ammonium nitrate and sulfate have been proved to exist in the freshly emitted biomass burning particles in many previous works (Silva et al., 1999; Li et al., 2015; Huo et al., 2016). In our work, we have seen ammonium nitrate and sulfate from single particle mass spectra as well, with peaks of $m/z$ 62 [$NO_3^-$], 97[$HSO_4^-$], and 18 [$NH_4^+$].

   Besides, we did not claim that ammonium nitrate and sulfate as the major components in biomass burning particles as the chemical information obtained from the single particle mass spectra is not quantitative.

2) the use of thermodenuding and bulk optical property measurement to infer things about coatings; I think there are some major assumptions that need to be addressed before confidently reaching this conclusion,

   **Response:** We are not quite sure about which conclusion the reviewer referred to, but we do have some technique issues about TD measurement to address in the response and the revised manuscript:

   **Particle loss in TD**
   In this work, we have taken the transport efficiency of TD into account for all the thermo-denuded related measurements including the absorption enhancement calculation. As written in Line 282, *"the measured η were used to correct the particle number concentration in the calculation of all the measurements related to thermal-denuded process"*.

   **Particle shrinkage**
   As for shrink factor, particles might shrink to smaller sizes after thermal treatment. In this experiment, we heated the particles before the DMA size selection. Thus, the mono-dispersed particles at the fixed size obtained after heating actually are not from the particles with the same original dried-particle diameter. For measurement like absorption enhancement, the particle optical data before and after thermal-denuded process are compared directly. Thus, it is essential to take the particle shrink factor into consideration.

   Here, we accepted the comments of the reviewers and added a new part (2.6) named as

"Shrink factor" (Line 285).

*"The thermal-denuded method to separate the coating of particles for absorption enhancement calculation as well as other experiments related to particle volatility has been used in previous work (Nakayama et al., 2014; Chan et al., 2011; Lack et al., 2012). However, the particles might shrink to smaller sizes after thermal treatment. The particle shrinkage should be taken into consideration for size-selected volatility experiments which was neglected in previous work. The major reason could be the extremely low concentration for size-selected particles after thermal-denuded process up to 300 $^o$C. The concentration of the size-selected particles was too low to be detected in the following instruments.*

*Therefore, we developed an approximation of the particle shrinkage calculation. A tandem DMAs (TDMA) was utilized to detect the size change of particles. Here, we used the ratio of the particle diameter after heating ($d_{m2}$) to the diameter before heating ($d_{m1}$) as the shrink factor ($d_{m2}/d_{m1}$) of particles (shown in Figure S3). An approximation of the peak value for the dominant shrink factor mode was used for each diameter. The selection of particle diameter after thermal-denuded process was based on the original dried-particle diameter multiplied the shrink factor of each diameter (discussed in supplementary)."*

3) the methods utilized to calculate effective densities are not adequately described,

**Response:** For the calculation of effective density, we have used a large space with pretty detailed information of the method in our manuscript (Line 171-233).

4) there is a body of literature from the FLAME experiments that I think could be used to better interpret these results,

**Response:** We accepted the reviewer's advice and added some discussion of our findings and the comparisons with previous studies performed by other groups.

Line 337: *Evidence of external mixing sodium and potassium salts in ambient environment was also observed by single particle mass spectrometry in previous work (Zauscher et al., 2013; Bi et al., 2011).*

Line 339: *A recent work performed by Lee et al. (2016) reported that $K^+$ was not uniformly mixed in biomass burning particles with less than 20% particles containing high $K^+$ content.*

Line 344: *The similar results of the externally mixed aerosol population was observed by Moffet et al. (2008) with a wide range of densities (1.1-3.4 g/ cm$^3$).*

Line 443: *The mass spectra of individual biomass burning particles have been studied in previous work (Silva et al., 1999; Zauscher et al., 2013).*

Line 572: *McMeeking et al. (2014) found that the strongly light-absorbing biomass burning particles tended to have a weak wavelength dependent absorption while the weakly light-absorbing particles tended have a strong wavelength dependent absorption, which is consistent with our results. In this work, the high values of AAE (~ 6.23) and SSA*

*(~0.89, at 530 nm) suggested the light absorbing of rice straw burning particles were relatively weak compared to the particles emitted from other types of biofuels.*

Line 585: *Previous studies have reported the absorption enhancement values in a range of 1.2-1.6 for biomass burning particles (Moffet and Prather, 2009; McMeeking et al., 2014). However, some other studies suggested that BC absorption enhancement due to lensing is minimal and climate models might overestimate the warming effect by BC (Healy et al., 2015; Cappa et al., 2012).*

Line 605: *Other than coating thickness, absorption enhancement of particles could be related with the mixing state and morphology (Liu et al., 2015).*

5) the composition measurements can be better analyzed and presented,

**Response:** We revised our manuscript based on the reviewer's advice. Please see the replies to detailed comments 15, 16, 23, 25 and 26.

6) the article needs to be better organized around a main point, and

**Response:** In this work, we focus on the size dependent mixing states of biomass burning particles and their correlations with the optical properties. To make this main point clear, we rewrote the abstract, revised some parts of the introduction and result discussions. Below is the new abstract:

*"Biomass burning aerosol has important impact on the global radiative budget. A better understanding of the correlations between the mixing states of biomass burning particles and their optical properties is the goal of a number of current studies. In this work, effective density, chemical composition, and optical properties of rice straw burning particles in the size range of 50-400 nm were measured using a suite of online methods. We found that the major components of rice straw burning particles included black carbon (BC), organic carbon (OC) and potassium salts, but the mixing states of particles were strongly size-dependent. Particles of 50 nm had the smallest effective density (1.16 $g/cm^3$), due to a relatively large proportion of aggregate BC. The average effective densities of 100-400 nm particles ranged from 1.35-1.51 $g/cm^3$ with OC and inorganic salts as dominant components. Both density distribution and single-particle mass spectrometry showed more complex mixing states in larger particles. Upon heating, the separation of the effective density distribution modes testified the external mixing state of less volatile BC or soot and potassium salts. Size-resolved optical properties of biomass burning particles were investigated at two wavelengths ($\lambda$=450 & 530 nm). The single scattering albedo (SSA) showed the lowest value for 50 nm particles (0.741 $\pm$0.007 & 0.889 $\pm$0.006) because of the larger proportion of BC content. Brown carbon played an important role for the SSA of 100-400 nm particles. The Ångström absorption exponent (AAE) values for all particles were above 1.6, indicating the significant presence of brown carbon in all sizes. Though freshly emitted, the light absorption enhancement ($E_{abs}$) was observed for particles larger than 200 nm because of the non-BC material coating. Concurrent measurements in our work provide a basis for discussing the physicochemical properties of biomass burning aerosol and its effects on global climate and atmospheric environment."*

7) I think the internal mixing of organics influence the effective density and this does not come through very well in the manuscript while I think it can be inferred with the available data.

**Response:** See the replies to the detailed comments 16 & 31.

In many places, the tables and figures can be described better and I have done my best to point that out. I think the article can be better proofread for grammar as well.

**Response:** We changed Fig. 3 and combined Fig. 3 and S3 together (detailed comment 22, 26). We revised the title of Table S1 (detailed comment 14).

In light of this critique, I recommend that the article undergo a major rewrite before it is acceptable for publication in ACP. The article will be a great addition to the literature after a significant transformation.

**Detailed Comments:**

1) L24: "relative" should be "relatively"

   **Response:** Changed.

2) L54: awkward sentence

   **Response:** We deleted this sentence.

3) L59: Surely the authors can provide more current refrences on BC – such as Tami Bond's extensive article published in JGR called "bounding black carbon".

   **Response:** We accepted the reviewer's advice and added this references in Line 50.

   *"BC, which is predominantly produced from the combustion related sources, absorbs solar radiation across the visible spectrum, resulting in a warming effect (Bond et al., 2013)".*

4) L60: awkward sentence

   **Response:** We deleted this sentence.

5) L70: suggest changing "low-visible" to "short wavelength visible"

   **Response:** Changed.

6) L89: The Tang an Munkelwitz article probably not the best reference here.

   **Response:** We accepted the reviewer's advice and changed the reference to Schmid et al., 2007.

   *Schmid, O., Karg, E., Hagen, D. E., Whitefield, P. D., and Ferron, G. A.: On the effective density of non-spherical particles as derived from combined measurements of aerodynamic and mobility equivalent size, J. Aerosol Sci., 38, 431-443, doi: 10.1016/j.jaerosci.2007.01.002, 2007.*

7) L105: This seems to be incorrect – bulk measurements cannot distinguish between

particles in a population.

**Response:** We changed sentence to *"distinctions among particles might be omitted by bulk measurements"*.

8) L114: grammar

**Response:** We rewrote the sentence as *"Biomass burning particle is a complex mixture of organic and inorganic species, including strongly light-absorbing BC and BrC. Size-resolved or even single particle level information on the morphology, chemical composition, and optical properties of biomass burning particles are necessary to have a better understanding of the correlations among these physiochemical properties."*.

9) L116: morphology of BB particles has been extensively documented via microscopy measurements. I think the introduction should include some of these. Most notably, Hopkins et al analyzed the optical and morphological properties of rice straw from the Flame experiment: Hopkins, R. J., K. Lewis, Y. Desyaterik, Z. Wang, A. V. Tivanski, W. P. Arnott, A. Laskin, and M. K. Gilles (2007), Correlations between optical, chemical and physical properties of biomass burn aerosols, Geophys. Res. Lett., 34, L18806, doi:10.1029/2007GL030502.

**Response:** We accepted the reviewer's advice and added the references in Line 77.

10) L119: Grammar

**Response:** We revised the sentence as *"In this study, laboratory experiments were conducted on rice straw combustion, a main source of biomass burning particles in Southern China"*.

11) L234: The procedures outlined in this section are not clear. Why are densities assumed to be unity? How will these assumptions be cancelled out later? What is being calculated and what procedures (e.g. iterative solutions…etc.) are being used?

**Response:** We listed the detailed procedures of the shape factor calculation a below:

$$\rho_p = \frac{m_p}{\frac{\pi}{6}d_{ve}^3} \tag{1}$$

$$\rho_{eff}^{I} = \frac{m_p}{\frac{\pi}{6}d_m^3} \tag{2}$$

$$\rho_{eff}^{II} = \frac{d_{va}}{d_m}\rho_0 \tag{3}$$

$$\frac{d_m}{C_c(d_m)} = \frac{d_{ve}\chi}{C_c(d_{ve})} \tag{4}$$

$$Cc(d) = 1 + \frac{2\lambda}{d}\left[\alpha + \beta\exp\left(-\gamma\frac{d}{2\lambda}\right)\right] \tag{5}$$

$$d_{va} = \frac{\rho_p}{\rho_0}\frac{d_{ve}}{\chi} \tag{6}$$

With assumed particle density ($\rho_p$) and known particle mass ($m_p$) measured by an APM, a calculated $d_{ve}$ could be obtained using Equation (1).

$$d_{ve} = \sqrt[3]{\frac{6m_p}{\pi\rho_p}} \tag{7}$$

Though $\rho_p$ was unknown, it would be canceled out later.

Using the same $d_{ve}$ and for any shape factor ($\chi$), a calculated $d_m$ and $d_{va}$ was obtained by Equation (4) and (6), respectively. Iterative solutions might be used in the calculation step of Equation (9)

$$\frac{d_m}{C_c(d_m)} = \frac{\sqrt[3]{\frac{6m_p}{\pi\rho_p}}\chi}{C_c(\sqrt[3]{\frac{6m_p}{\pi\rho_p}})} \tag{8}$$

$$Cc\left(\sqrt[3]{\frac{6m_p}{\pi\rho_p}}d\right) = 1 + \frac{2\lambda}{\sqrt[3]{\frac{6m_p}{\pi\rho_p}}}[\alpha + \beta\exp(-\gamma\frac{\sqrt[3]{\frac{6m_p}{\pi\rho_p}}}{2\lambda})] \tag{9}$$

$$d_{va} = \frac{\rho_p}{\rho_0}\frac{\sqrt[3]{\frac{6m_p}{\pi\rho_p}}}{\chi} \tag{10}$$

$\lambda$ is the mean free path of gas molecules. The empirical constants $\alpha$, $\beta$, and $\gamma$ are 1.142, 0.558, and 0.999 respectively (Allen and Raabe, 1985).

We could simplify Equation (8) and (10) into $d_m = A*m_p*\chi$, and $d_{va} = B*m_p/\chi$. A and B were simplified coefficients in the calculation.

Thus, $\rho_{eff}^{II}$ could be obtained by the calculated $d_m$ and $d_{va}$ and an estimated $m_p$ was calculated by replacing $\rho_{eff}^{I}$ by $\rho_{eff}^{II}$ in Equation (2).

$$\rho_{eff}^{II} = \frac{d_{va}}{d_m}\rho_0 = \frac{B}{A\chi^2} = \frac{m_p}{\frac{\pi}{6}d_m^3} \tag{11}$$

$$m_{p-estimated} = \frac{\pi}{6}\frac{B}{A}\frac{d_m^3}{\chi^2} \tag{12}$$

We then calculated the ratio of the estimated $m_p$ to the exact $m_p$ as a function of $d_m$ and $\chi$ (shown in Figure S5, discussed in Section 3.1.5).

We have changed the sentence into "*though $\rho_p$ was unknown, nevertheless it would be canceled out later*".

12) L241: This title is too general to be of any use.

**Response:** We changed this title to *"Instrumentation for effective density measurements"*.

13) L249: change to Kr-85 (use a dash)

**Response:** We accepted the reviewer's advice and changed to "Kr-85".

14) L297 and Table S1: Xc1 and Xc2 need to be defined - I assume these are some indication

the peak positions. With-out knowing this it is difficult to make sense of the table. How did the peak positions change for the different fits? What do the numbers in the Table S1 indicate? Area? Height?

**Response:** $x_{c1}$ and $x_{c2}$ are the peak values of the effective density modes in Fig. 2.

We accepted the reviewer's advice and changed "$x_{c1}$" and "$x_{c2}$" to "$Eff_1$" and "$Eff_2$" in Table S1.

*Table S1. Peak values and R squares of the average density distribution of 50, 100, 200, and 400 nm particles at 20 $^{o}C$ (room temperature), 150$^{o}C$, and 300$^{o}C$. $Eff_1$ and $Eff_2$ are the peak values of the effective density modes in Fig. 2. $R^2$ is the square of the correlation coefficient using Gaussian model fitting.*

| | 50 nm | | | 100 nm | | | 200 nm | | | 400 nm | | |
|---|---|---|---|---|---|---|---|---|---|---|---|---|
| | $Eff_1$ | $Eff_2$ | $R^2$ | $Eff_1$ | $Eff_2$ | $R^2$ | $Eff_1$ | $Eff_2$ | $R^2$ | $Eff_1$ | $Eff_2$ | $R^2$ |
| 20$^{o}$C | 1.167 | \ | 0.980 | 0.933 | 1.454 | 0.979 | 0.939 | 1.519 | 0.9697 | 1.344 | 1.917 | 0.966 |
| 150$^{o}$C | 0.972 | 1.642 | 0.951 | 0.981 | 1.691 | 0.932 | 0.984 | 1.746 | 0.949 | 1.094 | 1.798 | 0.911 |
| 300$^{o}$C | 0.976 | 1.756 | 0.851 | 0.994 | 1.851 | 0.864 | 1.030 | 1.857 | 0.850 | 1.157 | 2.051 | 0.779 |

15) L312-317: What temperature is being referred to? I think the density profile for the room temperature particles is NOT bimodal. The data for (what is presumably) the second mode is noisy.

**Response:** The temperature in Line 312-317 was room temperature (we added this information in Line 325). The density distribution discussed here was for particles at 100 nm at room temperature. The shoulder peak at ~1.0 g/cm$^3$ was weak but quite obvious. The existence of this peak as a separate mode had been proved by the thermal desorption experiment as shown in Fig. 2.

Furthermore, figure S1 seems to have some fit parameters that need to be more fully described.

**Response:** Figure S1 displays the average number size distribution detected by SMPS. No fitting parameter was applied. We added "*detected by the Scanning Mobility Particle Sizer (SMPS)*" in the figure caption of Figure S1.

The bimodal gaussian fits should be used in the discussion. To state that the two components are externally mixed after passing through the thermodenuder is misleading.

**Response:** We did use the bimodal Gaussian distribution to fit the density modes shown as the dash lines in Figure 2. Here, we only discussed the room temperature measurement. The appearance of the peak at ~1.0 g/cm$^3$ suggested that BC was partly externally mixed with other components in biomass burning particles.

16) L327: Whos to say that internally mixed OC and potassium salts do not contribute to modes with lower densities? In fact, I think data shown in Figure 3 and S3 support the internal mixing of potassium and organics. The mass spectra shown in Figure S6 also support this. Its not clear why they did not look at the chemical changes as a function of temperature.

**Response:** We agree with this comment. We reword the statement as following:

*"The dominant modes for biomass burning particles in the size range of 50-400 nm (Figure 2) could be a mixture of similar composition (BC, OC, potassium salts and secondary inorganic species) but different proportions."*

17) L336: Organic matter can be secondary, so the parenthetic statement is inaccurate.

**Response:** We accepted the reviewer's advice and revised in the manuscript as *"BC, OC, potassium salts and secondary inorganic species"*.

18) L349: What is "deviation range"? It would be useful to mark out suspected doubly charged modes on the figure.

**Response:** To verify the existence of doubly or multiply charged particles, the equation

$$\frac{d_m}{C} = \frac{2neVL}{3\mu q_{sh} ln\left(\frac{r_2}{r_1}\right)}$$

is used to calculate the particle diameter ($d_m$) for particles with different number of charges ($n$) at a set DMA voltage ($V$), DMA rod length ($L$), gas viscosity ($\mu$), sheath flow ($q_{sh}$), inner radius of the DMA annular space ($r_1$), and outer radius of the DMA annular space ($r_2$). $C$ is the Cunningham slip factor evaluated at $d_p$ (Knutson and Whitby, 1975).

Thus, the diameters of doubly charged particles should be larger than $d_m = \frac{4eVLC}{3\mu q_{sh} ln\left(\frac{r_2}{r_1}\right)}$.

We deleted the "deviation" to avoid any misleading.

Since the doubly charged particle is not the main topic or finding in our work, we did not show the calculation method or mark the doubly charged mode in the manuscript.

19) L356: "bi-model" should be "bimodal".

**Response:** Changed.

20) L392: Why is one method for obtaining "effective density" consistently lower than the other? Can other information be extracted from this? I dont understand the value of reporting results from the two methods.

**Response:** Overall, these two methods had consistent results. The differences between the average values from the two methods were less than 8% for all particle sizes. We noticed that $\rho_{eff}^{II}$ were generally smaller than $\rho_{eff}^{I}$, which could be due to the systematic error from different measurements.

We added the above discussion in our manuscript.

We didn't mean to use two methods to measure the effective density in this work. Since the SPAMS could obtain the aerodynamic diameters simultaneously, we determined to present results from both calculations.

21) L434: I highly doubt the material density of the BB-KCl type is in the CRC handbook – in fact the CRC is referenced repeatedly in weird places.

**Response:** Here, ~1.99 g/cm$^3$ is the material density of KCl we found from CRC handbook (Lide, 2008). We revised the sentence to "*For example, the BB-KCl type might have higher effective density compared with others since the dominant composition KCl has a material density of ~1.99 g/cm$^3$ (Lide, 2008).*"

The CRC was referenced three times in our paper (Line 336, 465, 473). The first two references were used to expound the material density of KCl and the third one was to tell the material density of $NH_4NO_3$ and $(NH_4)_2SO_4$.

22) L436: Which is the "first" and which is the "second" mode?

**Response:** We accepted the reviewer's advice and numbered each mode in Fig. 3.

[Figure]

23) L442: I think this should be better referenced. What data suggests ammonium nitrate and sulfate are the dominant composition from rice straw burning? I dont think the measured density is a reliable way to infer chemical composition due to the fact that the material density may be a weighted average of organic and inorganic species. If ammonium salts are so prevalent, one would expect m/z = 18 to be fairly prominent - this is not mentioned in the manuscript.

**Response:** Firstly, we did not claim that ammonium nitrate and sulfate as the major components in biomass burning particles since the chemical information obtained from the single particle mass spectrometry is not quantitative. Ammonium nitrate and sulfate have been proved to exist in the freshly emitted biomass burning particles in previous works (Silva et al., 1999; Li et al., 2015; Huo et al., 2016). In our work, we have seen ammonium nitrate and sulfate from single particle mass spectra as well, with peaks of *m/z* 62 [$NO_3^-$], 97[$HSO_4^-$], and 18 [$NH_4^+$].

The peak area of each chemical species in single particle mass spectra is mostly determined by its ionization cross section in the laser ablation. The low ion intensity of *m/z* 18 [$NH_4^+$] is caused by the low ionization cross section of ammonium in the 266 nm laser ablation.

We agree that the particle density may be a weighted average of its organic and inorganic components. Here, we didn't infer the chemical compositions purely based on the density measurement. In this part of the discussion, we tried to combine the mass spectrometry and density measurements to see the connections between the density modes and the mixing of major particle components.

24) L452: Is ammonium nitrate really expected to be amorphous? This needs to be proven (with references), otherwise it is very speculative.

**Response:** We added a reference in Line 483 to confirm this statement.

Audebrand, N., Auffredic, J. P., and Louer, D.: Thermal decomposition of cerous ammonium nitrate tetrahydrate studied with temperature-dependent X-ray powder diffraction and thermal analysis, Thermochim. Acta, 293, 65-76, doi: 10.1016/s0040-6031(97)00064-6, 1997.

25) L457: these modes are hardly discernable – especially the middle mode in figure 3b.

**Response:** We agree with this comment. The first two modes of 400 nm particles at 20 $^{o}$C were quite close and their chemical compositions were also similar. We assume these two modes were derived from one mode. We revised the related discussion in Line 488.

*"For 400 nm mobility selected particles, the pie charts of particle type were almost identical for the first and second modes (as shown in Fig. 3b, 20 $^{o}$C). Thus, we assume these two modes were derived from one effective density mode."*

26) L464-472: I think the differences in SPMS data at the different temperatures is very telling. The drastic decrease in the OC cluster at high temperatures can explain why the effective density increases: these particles loose low density organics and become more dense. I think this point is missing from the discussion of the densities. The loss of organics and other low volatility secondary species is not suprising and can be shown better thorugh the use of difference spectra.

**Response:** We accepted this suggestion and added this point in Line 501.

*"The high effective density (>2.0) of biomass burning particles at 300$^{o}$C could be due to the vaporization of volatile organics with low density since the BB-OC type decreased drastically after thermal treatment."*

The mass spectra of biomass burning particles before and after thermal treatment could be found in our previous work (Zhai et al., 2015). Here, we used the pie charts of particle type to illustrate the changes before and after thermal-denuded process, which is more drastic in terms of the decrease of OC proportion (BB-OC type) comparing with the difference mass spectra.

Furthermore, I do not see why Figures 3 and S3 are separate. I think it would be much more impactful to show these figures together.

**Response:** We accepted this advice and combined Fig. 3 and S3 into one figure.

27) L487: what is offset here? Forcing? This is unclear.

**Response:** We rewrote the sentence as "*It's worth noting that the optical measurement was based on bulk measurement by CAPSs, which is not sensitive to the diversity of particle mixing state.*"

L560: This sentence is unclear. Furthermore, the procedures referred to in section 2.5 are not really described in that section, so I suggest that the authors give this some detail.

**Response:** Please see the response to the general comment 2. We added a new part (2.6) named as "Shrink factor" in Line 285 to give the detailed description.

Another more general comment about using thermodenuders to estimate absorption enhancement: how might the TD cause side reactions to affect optical properties?

**Response:** In our experiment, we used nitrogen gas (99.999%) to dilute the biomass burning smoke in our chamber with a dilution ratio ~ 6. The nitrogen environment could help to avoid the side reactions like thermal oxidation.

28) L567: The above definition (L 553) of absorption enhancement is really a bulk definition. Without knowledge of the exact mixing state of the population, I think it is difficult to attribute "absorption enhancement" to mixing state effects, no?

**Response:** Indeed the particle absorption enhancement is a bulk definition. Here, the discussion of mixing state is based on the assumption of core-shell structure of particles which has been widely applied in previous work (Cappa et al., 2012; Lack et al., 2012; Nakayama et al., 2014). The non-BC coating material can enhance light absorption of BC core through "lensing effect". A thicker coating, which equals to a thicker lensing, can induce a larger absorption enhancement.

In this work, we didn't use the single particle data to over-interpret the absorption enhancement results.

29) L589: "nonspherical" or "fractal" may be a better term to use in place of "aggregate".

**Response:** We accepted the suggestion and changed "aggregate" to "fractal".

30) L606: again I think lower densities may be caused by mixing with organics – as described above.

**Response:** We agree with this comment and add "low density organics" in the sentence.

31) L613: "acceptable standard" does not seem right here. What is "typical"?

**Response:** We changed our wording to "*indicating the significant presence of brown carbon in all sizes*" (Line 651).

32) L616: change "volatile" to "volatility"

**Response:** Changed.

33) L619: How can the authors attribute absorption enhancement to coating thickness using bulk measurements? Here the conclusions do not really follow from the data.

**Response:** See the response to comment 28.

**References:**

Allen, M. D., and Raabe, O. G.: Slip correction measurements of spheical solid aerosol-particles in an improved millikan apparatus, Aerosol Sci. Technol., 4, 269-286, 10.1080/02786828508959055, 1985.

Cappa, C. D., Onasch, T. B., Massoli, P., Worsnop, D. R., Bates, T. S., Cross, E. S., Davidovits, P., Hakala, J., Hayden, K. L., Jobson, B. T., Kolesar, K. R., Lack, D. A., Lerner, B. M., Li, S. M., Mellon, D., Nuaaman, I., Olfert, J. S., Petaja, T., Quinn, P. K., Song, C., Subramanian, R., Williams, E. J., and Zaveri, R. A.: Radiative Absorption Enhancements Due to the Mixing State of Atmospheric Black Carbon, Science, 337, 1078-1081, 10.1126/science.1223447, 2012.

Huo, J., Lu, X., Wang, X., Chen, H., Ye, X., Gao, S., Gross, D. S., Chen, J., and Yang, X.: Online single particle analysis of chemical composition and mixing state of crop straw burning particles: from laboratory study to field measurement, Frontiers of Environmental Science & Engineering, 10, 244-252, 10.1007/s11783-015-0768-z, 2016.

Knutson, E. O., and Whitby, K. T.: Aerosol classification by electric mobility: apparatus, theory, and applications, Journal of Aerosol Science, 6, 443-451, 1975.

Lack, D. A., Langridge, J. M., Bahreini, R., Cappa, C. D., Middlebrook, A. M., and Schwarz, J. P.: Brown carbon and internal mixing in biomass burning particles, Proceedings of the National Academy of Sciences of the United States of America, 109, 14802-14807, 10.1073/pnas.1206575109, 2012.

Li, C., Ma, Z., Chen, J., Wang, X., Ye, X., Wang, L., Yang, X., Kan, H., Donaldson, D. J., and Mellouki, A.: Evolution of biomass burning smoke particles in the dark, Atmos. Environ., 120, 244-252, 10.1016/j.atmosenv.2015.09.003, 2015.

Nakayama, T., Ikeda, Y., Sawada, Y., Setoguchi, Y., Ogawa, S., Kawana, K., Mochida, M., Ikemori, F., Matsumoto, K., and Matsumi, Y.: Properties of light-absorbing aerosols in the Nagoya urban area, Japan, in August 2011 and January 2012: Contributions of brown carbon and lensing effect, Journal of Geophysical Research-Atmospheres, 119, 12721-12739, 10.1002/2014jd021744, 2014.

Silva, P. J., Liu, D. Y., Noble, C. A., and Prather, K. A.: Size and chemical characterization of individual particles resulting from biomass burning of local Southern California species, Environmental Science & Technology, 33, 3068-3076, 10.1021/es980544p, 1999.

Zhai, J., Wang, X., Li, J., Xu, T., Chen, H., Yang, X., and Chen, J.: Thermal desorption single particle mass spectrometry of ambient aerosol in Shanghai, Atmos. Environ., 123, 407-414, 10.1016/j.atmosenv.2015.09.001, 2015.

---

## Author Comment (AC2) · 14 Mar 2017

**Response to Review 2**

We sincerely thank the reviewer for the valuable comments and suggestions. Below we list our point-by-point replies to the comments and the descriptions of the changes we made in the revised manuscript.

Zhai and coauthors describe a series of experiments used to investigate the chemical, physical and optical properties of particles generated through the combustion of rice straw, chosen to represent local crop residues in Southern China. The motivation is to provide data that will help modelers to better assess regional and global radiative climate impacts of particles from this source. A combination of instrumentation is applied to calculate effective densities in two ways, employing a scanning mobility particle sizer, single particle mass spectrometer and an aerosol particle mass analyzer. Extinction, scattering and calculated absorption coefficients are also derived from a cavity attenuated phase shift spectrometer. These properties are assessed for untreated and thermally denuded rice straw combustion particles. While the experiments are worthwhile, there are some issues regarding the approaches taken, and the results need to be framed in the context of existing work in a more rigorous way. Suggested revisions are below:

**Major comments:**

1) Why is the rice straw dried at 100 ℃ prior to use? Presumably local farmers do not do this and simply burn the residue with its natural water content. Wouldn't this change the burn conditions, smouldering etc? Would it be more representative to burn untreated fuel?

   **Response:** Water content in crop residues varies in a wide range and has a great impact on the burning efficiency and emissions (Hayashi et al., 2014; Oanh et al., 2011). Crop residue moisture has been shown to be positively correlated with particle emissions in the range of 5~35 wt. %. However, empirical emission inventory calculation has to simplify the variation from the water content in biofuels. Pretreatment of biomass fuel in burning simulation is a practical and necessary procedure to ensure the result can be comparable with other studies under the defined conditions (e.g. dehydration at 100 $^{\circ}$C for 24 h to ensure water content of the residue within 2 wt. %), which has been applied in many previous studies (Hayashi et al., 2014; Li et al., 2015; Oanh et al., 2011; Zhang et al., 2011).

   In this work, water content in the rice straw we collected from the field was less than 5 wt. %. The dehydration to reach a water content within 2 wt. % had not much influence on the burning emissions.

2) Why is a relatively low laser desorption ionization energy (0.6 mJ per pulse) selected here? Routinely single particle mass spectrometers of this design employ energies on the order of 1 mJ per pulse. How were the ART-2a cluster parameters chosen- are they simply selected from previous work? If so cite the source and reason for selection.

   **Response:** In this work, laser energy for the desorption/ionization could be varied between 0.5 and 1 mJ with a spot size of ~ 0.3 mm. Higher laser power could yield more elemental compositions and fragments of organics. Lower laser fluencies could yield a greater amount of molecular information. 0.6 mJ/pulse is a proper laser

desorption/ionization energy for SPAMS. The laser power density of SPAMS (0.6 mJ with a spot size of 0.3 mm) is comparable to that of ATOFMS (~1.0 mJ with a spot size of ~1.0 mm).

When using ART-2a method, the learning rate could affect the rate of convergence which will result in different amount of clusters. Generally, with the same vigilance, a higher learning rate could result in a greater amount of clusters. The vigilance could also influence the amount of clusters. At low vigilance, the result of ART-2a is not accurate with rough resolution. At high vigilance, too many clusters cloud be generated with vague features of mass pattern. The ART-2a parameters used in this paper were based on previous work and the experience in our group (Song et al., 1999; Huang et al., 2013; Spencer et al., 2007).

We have added the references in Line 165 as *"Base on previous work (Huang et al., 2013; Spencer et al., 2007), parameters for ART-2a used in this work such as vigilance factor, learning rate, and iterations were 0.85, 0.05, and 20, respectively"*.

3) Is a flow rate of 0.6 L min-1 within the manufacturer's operating range for the thermal denuder?

**Response:** Yes. According to the manufacturer's specification, the flow range of thermodenuder (Model 3065, TSI Inc.) is 0.2 to 2 L/min with optimal flow at 0.5 to 1 L/min (http://www.tsi.com/Low-Flow-Thermodenuder-3065).

4) The transmission efficiency of the thermal denuder was tested using NaCl particles. Why not simply examine the transmission efficiency of size selected biomass combustion particles at room temperature to validate agreement with the NaCl transmission efficiencies at room temperature.

**Response:** In general, particle loss in any TD is caused by diffusional and thermophoretic processes, which are both size- and temperature-dependent (Wehner, et al., 2002). Thus, the loss measurements for a series of particle diameter and heater temperatures are essential.

Since sodium chloride (NaCl) is easy to generate and evaporates only at 600 ◦C, it was used to detect the transmission efficiency and particle loss of thermodenuder in our work and many previous work. Similar tests have been done to detect transmission efficiency in previous work (Huffman et al., 2008; Cheung et al., 2016).

**Also**, for the absorption enhancement calculation, the transmission efficiency needs to be taken into account. The formula on line 553 simply relates $b_{abs}$ before and after the denuder. Any losses of BC through diffusion and impaction in the denuder will be incorrectly assigned as absorption reduction through a removal of coating materials (absorption enhancement effect).

**Furthermore**, why is the "shrink factor" used to account for volatilized coating materials? Firstly, the shrink factors shown in Fig S7 demonstrate a wide range of values, with multiple modes, and using a single value does not seem appropriate. In any case accounting for shrinking should not be necessary. Shrinking of the coatings is desirable for this test. The idea is that the bulk BC mass should be conserved through the denuder

(applying the correct transmission efficiency factor should result in BC mass entering the denuder being roughly equivalent to "transmission corrected" BC mass exiting the denuder), but that the coating materials should evaporate. Any reduction in absorption can thus be assigned to either reduced optical "lensing" due to the reduction of coating sizes or evaporation of any non-BC absorbing material. Applying the transmission efficiency correction and removing the shrink factor should be tested.

**Response:** In our work, we have taken the transport efficiency of TD into account for all the thermo-denuded related measurements including the absorption enhancement calculation. As written in Line 282, *"the measured $\eta$ were used to correct the particle number concentration in the calculation of all the measurements related to thermal-denuded process"*.

As for shrink factor, particles might shrink to smaller sizes after thermal treatment. In this experiment, we heated the particles before the DMA size selection. Thus, the mono-dispersed particles at the fixed size obtained after heating actually were not from the particles with the same original dried-particle diameter. For measurement like absorption enhancement, the particle optical data before and after thermal-denuded process are compared directly. Thus, it is essential to take the particle shrink factor into consideration.

Here, we accepted the comment of Reviewer 3 and we added a new part (2.6) named as shrink factor (Line 285) to give a detailed description on the shrinking effect.

*"The thermal-denuded method to separate the coating of particles for absorption enhancement calculation as well as other experiments related to particle volatility has been used in previous work (Nakayama et al., 2014; Chan et al., 2011; Lack et al., 2012). However, the particles might shrink to smaller sizes after thermal treatment. The particle shrinkage should be taken into consideration for size-selected volatility experiments which could be neglected in previous work. One main reason should be due to the extremely low concentration for size-selected particles after thermal-denuded process up to $300\,^{o}C$. The concentration of the size-selected particles might be too low to be detected in the following instruments.*

*Therefore, we developed an approximation of the particle shrinkage calculation. A tandem DMAs (TDMA) was utilized to detect the size change of particles. Here, we used the ratio of the particle diameter after heating ($d_{m2}$) to the diameter before heating ($d_{m1}$) as the shrink factor ($d_{m2}/d_{m1}$) of particles (shown in Figure S3). An approximation of the peak value for the dominant shrink factor mode was used for each diameter. The selection of particle diameter after thermal-denuded process was based on the original dried-particle diameter multiplied the shrink factor of each diameter (discussed in supplementary). "*

5) The x-axis of Figure 3 should show include effective density also. It would also be very useful to show an effective density distribution plot for each of the different particle classes. This will allow the authors to provide better evidence to support their postulations regarding potential composition-density relationships throughout.

**Response:** We accepted this suggestion and the suggestion in minor comment 1. We

changed our figure as following and combined the original Fig. 3 and S3 into one figure.

[Figure]

6) Overall, there needs to be much better discussion of the findings here in the context of similar work previously performed by other groups. In terms of single particle mass spectrometry analyses of biomass burning particles (and relating biomass particle composition to optical and physical properties) there is a host of relevant studies that should be discussed. See (Silva et al., 1999; Zauscher et al., 2013; Bi et al., 2011; Guazzotti et al., 2003; Schwarz et al., 2008; Moffet and Prather, 2009; Moffet et al., 2008; Pagels et al., 2013). In terms of efforts examining relationships between biomass burning chemical composition measured through mass spectrometry and optical absorption enhancements see (McMeeking et al., 2014; Healy et al., 2015) and other ambient combustion particle absorption enhancements (Cappa et al., 2012; Liu et al., 2015) among others. One point which is posed as a new conclusion (evidence for external mixing of K and organic aerosol in biomass burning particles using single particle mass spectrometry) has been recently reported by others (Lee et al., 2016).

**Response:** We accepted the reviewer's advice and added discussion of the findings of similar work previously as the reviewer suggested.

Line 337: *Evidence of external mixing sodium and potassium salts in ambient environment was also observed by single particle mass spectrometry in previous work (Zauscher et al., 2013; Bi et al., 2011).*

Line 339: *A recent work performed by Lee et al. (2016) reported that $K^+$ was not uniformly mixed in biomass burning particles with less than 20% particles containing high $K^+$ content.*

Line 344: *The similar results of the externally mixed aerosol population have been found by Moffet et al. (2008) with an observation of a wide range of densities (1.1-3.4 g/ $cm^3$).*

Line 443: *The mass spectra of individual biomass burning particles have been studied in previous work (Silva et al., 1999; Zauscher et al., 2013).*

Line 572: *McMeeking et al. (2014) found that the strongly light-absorbing biomass burning particles tended to have a weak wavelength dependent absorption while the weakly light-absorbing particles tended have a strong wavelength dependent absorption, which is consistent with our results. In this work, the high values of AAE (~ 6.23) and SSA*

*(~0.89, at 530 nm) suggested the light absorbing of rice straw burning particles were relatively weak compared to the particles emitted from other types of biofuels.*

Line 585: *Previous studies have reported the absorption enhancement values in a range of 1.2-1.6 for biomass burning particles (Moffet and Prather, 2009; McMeeking et al., 2014). However, some other studies suggested that BC absorption enhancement due to lensing is minimal and climate models might overestimate the warming effect by BC (Healy et al., 2015; Cappa et al., 2012).*

Line 605: *Other than coating thickness, absorption enhancement of particles could be related with the mixing state and morphology (Liu et al., 2015).*

**Minor comments:**

1) Some SI figures should be in the main manuscript (S3 and S6)

   **Response:** We have accepted the reviewer's advice and combined Fig. 3 and S3 into one figure.

   However, the mass pattern of each cluster from biomass burning particles has been reported in previous work (Huo et al., 2016; Zauscher et al., 2013). The single particle mass pattern of biomass burning particles is not the major finding in this work. Thus, we would rather put Fig. S6 in the supplementary.

2) Figure S4: the legend does not specify the method used for effective density calculation

   **Response:** The description of different methods for effective density calculation was given in the figure caption of Fig. S4.

   *"Size-resolved effective density of biomass burning particles determined by two methods. $\rho_{eff}^{I}$ is the effective density obtained from mobility and mass measurements (based on the DMA-APM-CPC system) while $\rho_{eff}^{II}$ is obtained from mobility and aerodynamic measurements (DMA-SPAMS system). The effective density of each size is the average peak value of the dominant mode from different scans. Error bars represent the standard deviations of the five replicate test results."*

   Section 3.2: Do all of the particle types need the prefix BB? All particles discussed are from biomass.

   **Response:** We used prefix BB for particle types to distinguish the subgroups of biomass burning particles from the other particle types in the ambient environment which were named as OC, EC, etc. for short in many previous works (Bi et al., 2011; Huang et al., 2013).

   Also, the order of the particle types does not match Fig S6

   **Response:** We changed the order of particle types in Fig. S6.

[Figure]

3) Line 392: Test for statistically significant differences

   **Response:** Overall, these two methods had consistent results. The differences between the average values from the two methods were less than 8% for all particle sizes. We noticed that $\rho_{eff}^{II}$ were generally smaller than $\rho_{eff}^{I}$, which could be due to the systematic error from different measurements.

   We added the above discussion in our manuscript.

4) Line 386: Simply remove the intermediate size from the plot if it is not represented in the other plot

   **Response:** The optimum sampling size range of SPAMS is 0.2-2.0 μm. To make the comparison of two methods in Fig. S4 meaningful, we would like to keep the data of 300 and 350 nm particles.

5) Line 453: Is there a way to demonstrate this? See suggestion of effective density plots for each particle class earlier

   **Response:** We accepted the reviewer's advice and changed Fig.3.

6) Line 487: This is not an offset, simply a bulk measurement.

   **Response:** We rewrote the sentence as "*It's worth noting that the optical measurement*

*was based on bulk measurement by CAPSs, which is not sensitive to the diversity of particle mixing state.*"

**References:**

Cheung, H. H. Y., Tan, H., Xu, H., Li, F., Wu, C., Yu, J. Z., and Chan, C. K.: Measurements of non-volatile aerosols with a VTDMA and their correlations with carbonaceous aerosols in Guangzhou, China, Atmospheric Chemistry and Physics, 16, 8431-8446, 10.5194/acp-16-8431-2016, 2016.

Hayashi, K., Ono, K., Kajiura, M., Sudo, S., Yonemura, S., Fushimi, A., Saitoh, K., Fujitani, Y., and Tanabe, K.: Trace gas and particle emissions from open burning of three cereal crop residues: Increase in residue moistness enhances emissions of carbon monoxide, methane, and particulate organic carbon, Atmos. Environ., 95, 36-44, 10.1016/j.atmosenv.2014.06.023, 2014.

Huang, Y., Li, L., Li, J., Wang, X., Chen, H., Chen, J., Yang, X., Gross, D., Wang, H., and Qiao, L.: A case study of the highly time-resolved evolution of aerosol chemical and optical properties in urban Shanghai, China, Atmos. Chem. Phys., 13, 3931-3944, 2013.

Huffman, J. A., Ziemann, P. J., Jayne, J. T., Worsnop, D. R., and Jimenez, J. L.: Development and Characterization of a Fast-Stepping/Scanning Thermodenuder for Chemically-Resolved Aerosol Volatility Measurements, Aerosol Sci. Technol., 42, 395-407, 10.1080/02786820802104981, 2008.

Huo, J., Lu, X., Wang, X., Chen, H., Ye, X., Gao, S., Gross, D. S., Chen, J., and Yang, X.: Online single particle analysis of chemical composition and mixing state of crop straw burning particles: from laboratory study to field measurement, Frontiers of Environmental Science & Engineering, 10, 244-252, 10.1007/s11783-015-0768-z, 2016.

Li, C., Ma, Z., Chen, J., Wang, X., Ye, X., Wang, L., Yang, X., Kan, H., Donaldson, D. J., and Mellouki, A.: Evolution of biomass burning smoke particles in the dark, Atmos. Environ., 120, 244-252, 10.1016/j.atmosenv.2015.09.003, 2015.

Oanh, N. T., Bich, T. L., Tipayarom, D., Manadhar, B. R., Prapat, P., Simpson, C. D., and Liu, L. J.: Characterization of Particulate Matter Emission from Open Burning of Rice Straw, Atmos. Environ., 45, 493-502, 10.1016/j.atmosenv.2010.09.023, 2011.

Song, X. H., Hopke, P. K., Fergenson, D. P., and Prather, K. A.: Classification of single particles analyzed by ATOFMS using an artificial neural network, ART-2A, Anal. Chem., 71, 860-865, 10.1021/ac9809682, 1999.

Spencer, M. T., Shields, L. G., and Prather, K. A.: Simultaneous measurement of the effective density and chemical composition of ambient aerosol particles, Environ. Sci. Technol., 41, 1303-1309, doi: 10.1021/es061425+, 2007.

Wehner, B., Philippin, S., and Wiedensohler, A.: Design and calibration of a thermodenuder with an improved heating unit to measure the size-dependent volatile fraction of aerosol particles, Journal of Aerosol Science, 33, 1087-1093, 10.1016/s0021-8502(02)00056-3, 2002.

Zauscher, M. D., Wang, Y., Moore, M. J. K., Gaston, C. J., and Prather, K. A.: Air Quality Impact and Physicochemical Aging of Biomass Burning Aerosols during the 2007 San Diego Wildfires, Environmental Science & Technology, 47, 7633-7643, 10.1021/es4004137, 2013.

Zhang, H., Hu, D., Chen, J., Ye, X., Wang, S. X., Hao, J. M., Wang, L., Zhang, R., and An, Z.:
   Particle size distribution and polycyclic aromatic hydrocarbons emissions from agricultural crop
   residue burning, Environ. Sci. Technol., 45, 5477-5482, 10.1021/es1037904, 2011.

---

## Author Comment (AC3) · 14 Mar 2017

**Response to Review 3**

We sincerely thank the reviewer for the valuable comments and suggestions. Below we list our point-by-point replies to the comments and the descriptions of the changes we made in the revised manuscript.

This manuscript reports the size-resolved chemical composition, effective density and optical properties of rice straw burning particles using different online instruments including DMA-APM-CPC, single particle aerosol mass spectrometer, and cavity attenuated phase shift (CAPS) spectroscopy.

First of all, the focus of this study is unclear and hence the significance and atmospheric implication should be explicitly highlighted in the abstract, introduction and conclusion.

**Response:** In this work, we focus on the size dependent mixing states of biomass burning particles and their correlations with the optical properties. To make this main point clear, we rewrote the abstract, revised some parts of the introduction and result discussions. Below is the new abstract:

*"Biomass burning aerosol has important impact on the global radiative budget. A better understanding of the correlations between the mixing states of biomass burning particles and their optical properties is the goal of a number of current studies. In this work, effective density, chemical composition, and optical properties of rice straw burning particles in the size range of 50-400 nm were measured using a suite of online methods. We found that the major components of rice straw burning particles included black carbon (BC), organic carbon (OC) and potassium salts, but the mixing states of particles were strongly size-dependent. Particles of 50 nm had the smallest effective density (1.16 g/cm³), due to a relatively large proportion of aggregate BC. The average effective densities of 100-400 nm particles ranged from 1.35-1.51 g/cm³ with OC and inorganic salts as dominant components. Both density distribution and single-particle mass spectrometry showed more complex mixing states in larger particles. Upon heating, the separation of the effective density distribution modes testified the external mixing state of less volatile BC or soot and potassium salts. Size-resolved optical properties of biomass burning particles were investigated at two wavelengths (λ=450 & 530 nm). The single scattering albedo (SSA) showed the lowest value for 50 nm particles (0.741±0.007 & 0.889±0.006) because of the larger proportion of BC content. Brown carbon played an important role for the SSA of 100-400 nm particles. The Ångström absorption exponent (AAE) values for all particles were above 1.6, indicating the significant presence of brown carbon in all sizes. Though freshly emitted, the light absorption enhancement (Eₐᵦₛ) was observed for particles larger than 200 nm because of the non-BC material coating. Concurrent measurements in our work provide a basis for discussing the physicochemical properties of biomass burning aerosol and its effects on global climate and atmospheric environment."*

Secondly, the results should be discussed in more detail. In particular, the observations regarding particle effective densities and single particle compositions should be better integrated in this manuscript in order to provide a more complete picture on the relationship of particle mixing state, morphology and effective density.

**Response:** We accepted most of the reviewer's suggestions. Please see the detailed responses to specific comment 3-5.

Lastly, there is an issue about the data quality in Section 3.3.3 (See specific comments) that should be addressed by the authors.

**Response:** We accepted most of the reviewer's suggestions. Please see the detailed responses to specific comment 1 and 6.

The manuscript should be proofread carefully before submission. Overall, I don't recommend this manuscript to be published in Atmospheric Chemistry and Physics in the current format. Below is the specific comments.

**Specific comments:**

1. Absorption enhancement results (Section 3.3.3 and Figure 5): Figure 1 demonstrated that aerosol particles were pre-treated by a thermodenuder before generating monodisperse particles using a DMA. The size-selected particles were then characterized by a few real-time instruments. Using such experimental setup, the size-resolved absorption enhancement factors reported in Figure 5 are actually not meaningful. The primary reason is that those enhancement factors were not determined by comparing particles with the same original dried-particle diameter. The whole particle size distribution should shift towards the lower particle size after thermal treatment (i.e. removal of coating materials). The original size of heated particles should be greater than that of particles without heating as shown in Figure S7. It is unclear how the shrink factor and transmission efficiency of particles presented in Section 3.3.3 can resolve this fundamental problem. Detail clarification is required to keep the related discussion in the manuscript.

   **Response:** In this work, we have taken the transmission efficiency of TD into account for all the thermo-denuded related measurements including the absorption enhancement calculation. As written in Line 282, "the measured $\eta$ were used to correct the particle number concentration in the calculation of all the measurements related to thermal-denuded process".

   As for shrink factor, particles might shrink to smaller sizes after thermal treatment. Thus, Thus, the mono-dispersed particles at the fixed size obtained after heating actually are not from the particles with the same original dried-particle diameter. For measurement like absorption enhancement, the particle optical data before and after thermal-denuded process are compared directly. Thus, it is essential to take the particle shrink factor into consideration.

   Here, we accepted the reviewer's advice and we added a new part (2.6) named as shrink factor (Line 285) to give a detailed description on the shrinking effect.

   *"The thermal-denuded method to separate the coating of particles for absorption enhancement calculation as well as other experiments related to particle volatility has been used in previous work (Nakayama et al., 2014; Chan et al., 2011; Lack et al., 2012). However, the particles might shrink to smaller sizes after thermal treatment. The particle shrinkage should be taken into consideration for size-selected volatility experiments*

*which could be neglected in previous work. One main reason should be due to the extremely low concentration for size-selected particles after thermal-denuded process up to 300 °C. The concentration of the size-selected particles might be too low to be detected in the following instruments.*

*Therefore, we developed an approximation of the particle shrinkage calculation. A tandem DMAs (TDMA) was utilized to detect the size change of particles. Here, we used the ratio of the particle diameter after heating ($d_{m2}$) to the diameter before heating ($d_{m1}$) as the shrink factor ($d_{m2}/d_{m1}$) of particles (shown in Figure S3). An approximation of the peak value for the dominant shrink factor mode was used for each diameter. The selection of particle diameter after thermal-denuded process was based on the original dried-particle diameter multiplied the shrink factor of each diameter (discussed in supplementary)."*

Furthermore, this manuscript emphasizes a few times that aerosol coating can act as a lens to enhance light absorption of aerosol particles in general. Nevertheless, the major conclusion of this work regarding light absorption enhancement (relative to pure BC) is due to the presence of atmospheric brown carbon from biomass burning emissions and without much discussion on the lensing effect. It is recommended to change the tone/wording in the text to avoid any potential confusion to the readers, especially for those are not familiar with this research topic.

**Response:** For absorption enhancement, the major point in this work is that absorption enhancement has been observed in freshly emitted biomass burning particles, compared to the numerous works on aged ambient aerosol. We believe that the absorption enhancement in this experiment was caused by the particle coating rather than the presence of brown carbon. In the manuscript, we claimed the contribution of the thick coating to the absorption enhancement for particles larger than 200 nm. We only attributed the stronger absorption enhancement at the wavelength of 450 nm to the presence of brown carbon when compared with the 530 nm data.

2. Figure 2 and Section 3.1.3: Similar to the comment #1, it is essential to highlight in the manuscript that the individual column in Figure 2 is not presenting results obtained from particles with the same original dried-particle diameter. The current writing is somewhat misleading. However, the findings observed from this figure is still useful even though direct comparison of plots displayed in the same column is inappropriate.

**Response:** We accepted the reviewer's advice and added the description in Line 385 as following:

*"It is worth noting that the thermal-denuded particle density distribution here was not from the particles with the same original dried-particle diameter. However, our observations are still meaningful since the evolution trends of density distribution after heating were similar despite of the particle size."*

3. To better organize the discussion of particle effective density, it is recommended to combine Section 3.1.2 and 3.1.4 in the manuscript.

**Response:** We accepted this suggestion and combined Section 3.1.2 and 3.1.4 together.

4. Effective density and chemical composition (Section 3.1.3):

1) Lines 358-360: Without quantitative chemical characterization in a single particle basis, it is hard to prove if some BC was externally mixed with other components based on the effective density measurement alone. BC particles with highly fractal structure and thin organic coating can lead to the similar observations, depending on the uncertainty of effective density measurement.

**Response:** We agree with the reviewer's point of "BC particles with highly fractal structure and thin organic coating could also lead to the low density mode in effective density distribution". In the revised manuscript, we softened our statement as:

*"The separation of the peaks after heating suggested that the some less volatile BC or soot with effective density of ~1.0 g/cm$^3$ was possibly externally mixed with other compositions."*

2) Lines 365-378: According to Figure 2, peaks are always observed at around 1.7-1.8 g/cm3 for the heated particles with diameter less than 400 nm. Please comment whether this observation is due to the presence of extremely low volatility organic aerosol materials generated. Furthermore, what are the vaporization temperature of KCl and other potassium salts.

**Response:** We accepted the reviewer's suggestion and added the important role the extremely low volatility organic compounds might play in the dominant effective density mode at 300 $^o$C in the revision.

The vaporization temperatures of KCl and other potassium salts are above 700 $^o$C (Knudsen et al., 2004), which we have added in our manuscript.

The revision in part 3.1.3 is as following (Line 407):

*"Besides, Bond et al. (2006) reported that the density of light-absorbing carbon should be 1.7-2.1 g/cm$^3$ which is quite high compared with the density of the volatile organics (0.61-0.90 g/cm$^3$). Saleh et al. (2014) had shown that the light-absorbing organics in biomass burning particles were extremely low volatility organic compounds. Thus, we assume these extremely low volatility organics should play an important role in the dominant effective density mode at 300$^o$C.*

*Upon heating, the density mode of KCl and partly K$_2$SO$_4$ at ~2.0 g/cm$^3$ was ambiguous as the dominant mode shifted right and overlapped with the KCl mode (dash lines shown in Figure 2). However, at 300 $^o$C, the dominant mode of 400 nm particles was at 2.05 g/cm$^3$ which fitted the density of potassium salts, indicating the main material of 400 nm heated (~800 nm unheated, detected by a tandem DMAs) biomass burning particles should be potassium salts with vaporization temperatures above 700$^o$C (Knudsen et al., 2004)."*

5. Section 3.2 and Figure 3: The authors may over-interpret their observations. The average chemical compositions are not sufficient enough (i.e. they are too similar) to explain the effective densities of the two particle modes presented in Figure 3a (200 nm). Similarly,

the first two particle modes in Figure 3b (400 nm) have the very similar average chemical compositions.

**Response:** In the revised manuscript, we changed Fig. 3 as below:

[Figure]

Different chemical composition proportions could be one of the reasons of different effective density of each mode. Other reasons could be due to the difference of particle morphology such as sphericity and shape factor. Here, we focused the aspect of chemical composition since this part (Section 3.2) was mainly based on the SPAMS cluster analysis.

In the pie charts of 200 nm particles at room temperature, the second mode with larger effective density could be due to less BB-EC and more BB-KCl and BB-CN. Thus, the effective density of the second mode is higher than the first one.

However, the chemical compositions were similar in the first and second modes of 400 nm particles at 20 °C. We assume these two modes were derived from one mode. We revised the related discussion in Line 488.

*"For 400 nm mobility selected particles, the pie charts of particle type were almost identical for the first and second modes (as shown in Fig. 3b, 20 °C). Thus, we assume these two modes were derived from one effective density mode."*

What are the particle number distributions of each cluster (lines 430-432)? Their particle number distributions should be able to extract from the results of cluster analysis. Estimation of effective densities of some clusters is possible with such additional information.

**Response:** Indeed, the number distributions of each particle cluster could be extracted from cluster analysis. However, the cluster analysis by SPAMS in terms of chemical compositions is a qualitative one. We cannot estimate the effective densities via absolute number distribution of each cluster. Besides, the morphology of particles in each cluster varied from one another which is difficult to give a quantitative estimation.

Furthermore, it is unclear how to separate the particle modes in Figure 3 for constructing the pie chart. Please provide sufficient detail.

**Response:** We used the smallest value between two modes as the separation point (as the red lines shown in the figure below).

[Figure]

6. Section 3.3.2, Lines 533-541: What is the particle size for data reported in Table S2?

**Response:** The data reported in Table S2 was the particles that freshly emitted from combustion of biomass. The size distribution was shown in Fig. S1 (~20-660 nm).

Significant drop of AAE for the thermodenuded particles are observed. Please comment on the significance and relative contributions of extremely low volatility brown carbon to the total light absorption properties of biomass burning aerosol observed in this study and compare their results with existing literature.

**Response:** The significant drop of AAE should be attributed to the large proportion of organics evaporated at 150-300 °C. However, at 300 °C, the AAE was still above 1.6, indicating the presence of the extremely low volatility brown carbon in biomass burning particles. We have revised the AAE part in our manuscript as following (Line 564):

*"The SSA and AAE values of total biomass burning particles are shown in Table S2. The decrease of SSA values upon heating was due to the vaporization of secondary inorganic species like NH₄NO₃ and less absorbing organics. The AAE values for all particles at 150 °C and 300°C were ~19% and ~64%, lower than those at room temperature (20 °C). The significant decrease of AAE at 300°C could be due to the vaporization of light-absorbing organics in the temperature range of 150-300 °C. However, the AAE value at 300°C was still above 1.6, indicating the presence of extremely low volatility light-absorbing organics in biomass burning particles. McMeeking et al. (2014) found that the strongly light-absorbing biomass burning particles tended to have a weak wavelength dependent absorption while the weakly light-absorbing particles tended to have a strong wavelength dependent absorption, which is consistent with our results. In this work, the high values of AAE (~ 6.23) and SSA (~0.89, at 530 nm) suggested the light absorbing of rice straw burning particles were relatively weak compared to the particles emitted from other types of biofuels."*

**Minor and technical comments:**

1. Please replace "bi-model" by "bi-modal" throughout the entire manuscript.

   **Response:** Changed.

2. Figure S4: The legend displays the same for both types of effective density. Please correct.

   **Response**: We redrew the Figure S4 as below:

[Figure]

3. Please use "low" and "high" to describe density.

   **Response:** Changed.

4. Line 198: Should it be "…increased as the size increased…"?

   **Response:** We revised the sentence as "the SSA ($\lambda$=450 nm) of 100-400 nm particles increased as the size increased" (Line 529).

---

## Author Response (AR2)

**Author's response:**

Dear Editor,

We sincerely thank you for the valuable comments. We accepted your and the referees' suggestion by removing the discussion on the light absorption enhancement (Section 3.3.3) and the shrink factor calculations (Section 2.6). All the related contents were also removed throughout the manuscript (abstract, introduction, conclusion and literature). The serial numbers of figures were changed as well. All the changes are marked in the marked-up version of the manuscript attached below.

Sincerely,

Xin YANG

Department of Environmental Science & Engineering

Fudan University

Handan Road, 200433

Shanghai, China

**A marked-up manuscript version**

[revised manuscript text omitted]

$Eff_2$ are the peak values of the effective density modes in Fig. 2. $R^2$ is the square of the
correlation coefficient using Gaussian model fitting.

|  | 50 nm | | | 100 nm | | | 200 nm | | | 400 nm | | |
|---|---|---|---|---|---|---|---|---|---|---|---|---|
|  | $Eff_1$ | $Eff_2$ | $R^2$ | $Eff_1$ | $Eff_2$ | $R^2$ | $Eff_1$ | $Eff_2$ | $R^2$ | $Eff_1$ | $Eff_2$ | $R^2$ |
| 20$^{\circ}$C | 1.167 | \ | 0.980 | 0.933 | 1.454 | 0.979 | 0.939 | 1.519 | 0.9697 | 1.344 | 1.917 | 0.966 |
| 150$^{\circ}$C | 0.972 | 1.642 | 0.951 | 0.981 | 1.691 | 0.932 | 0.984 | 1.746 | 0.949 | 1.094 | 1.798 | 0.911 |
| 300$^{\circ}$C | 0.976 | 1.756 | 0.851 | 0.994 | 1.851 | 0.864 | 1.030 | 1.857 | 0.850 | 1.157 | 2.051 | 0.779 |

Table S2. Single scattering albedo (SSA) at wavelengths of 530 nm and 450 nm and
Ångström absorption exponent (AAE) of total biomass burning particles at 20 $^{\circ}$C
(room temperature), 150 $^{\circ}$C, and 300 $^{\circ}$C.

| Temperature | SSA | | AAE |
|---|---|---|---|
|  | 450 nm | 530 nm |  |
| Room temperature (20 $^{\circ}$C) | 0.750 | 0.897 | 6.230$\pm$0.160 |
| 150 $^{\circ}$C | 0.533 | 0.723 | 5.047$\pm$0.246 |
| 300 $^{\circ}$C | 0.469 | 0.560 | 2.229$\pm$0.292 |

[Figure]

Figure S1. Average number size distribution of biomass burning particles detected by
the Scanning Mobility Particle Sizer (SMPS).

[Figure]

Figure S2. Transport efficiency of NaCl in the thermodenuder as a function of particle diameter and heating temperature.

[Figure]

Figure S3. The size change ($d_{m2}/d_{m1}$) for particles in the size range of 50-400 nm when heated at 300 °C.

An approximation of the peak value for the dominant shrink factor mode was used for each diameter. For the original dried-particle diameter at 50, 100, 200, and 400 nm, the approximate shrink factor values are 0.99, 0.78, 0.61, and 0.60, respectively. Thus, particles at 50, 80, 120, and 240 nm were selected after thermal-denuded process for optical properties calculation.

[Figure]

Figure S34. Size-resolved effective density of biomass burning particles determined by two methods. $\rho_{eff}^{I}$ is the effective density obtained from mobility and mass measurements (based on the DMA-APM-CPC system) while $\rho_{eff}^{II}$ is obtained from mobility and aerodynamic measurements (DMA-SPAMS system). The effective density of each size is the average peak value of the dominant mode from different scans. Error bars represent the standard deviations of the five replicate test results.

[Figure]

Figure S45. Contour plot of the ratio of the estimated particle mass ($m_p$) to the exact $m_p$. The estimated $m_p$ was obtain by replacing $\rho_{eff}^{I}$ with calculated $\rho_{eff}^{II}$ in Equation (2). The ratios of the estimated $m_p$ by replacing $\rho_{eff}^{I}$ with exact $\rho_{eff}^{II}$ in Equation (2) to the exact $m_p$ was shown as well (red dots).

[Figure]

Figure S56. Average mass spectra of 6 particle types classified from biomass burning
particles.